# Rectified Diffusion Guidance for Conditional Generation

## Abstract

Classifier-Free Guidance (CFG), which combines the conditional and uncondi-
tional score functions with two coefficients summing to one, serves as a practical
technique for diffusion model sampling. Theoretically, however, denoising
with CFG *cannot* be expressed as a reciprocal diffusion process, which may
consequently leave some hidden risks during use. In this work, we revisit the
theory behind CFG and rigorously confirm that the improper configuration of the
combination coefficients (*i.e.*, the widely used summing-to-one version) brings
about expectation shift of the generative distribution. To rectify this issue, we
propose ReCFG[1] with a relaxation on the guidance coefficients such that denoising
with ReCFG strictly aligns with the diffusion theory. We further show that our
approach enjoys a ***closed-form*** solution given the guidance strength. That way, the
rectified coefficients can be readily pre-computed via traversing the observed data,
leaving the sampling speed barely affected. Empirical evidence on real-world
data demonstrate the compatibility of our post-hoc design with existing state-
of-the-art diffusion models, including both class-conditioned ones (*e.g.*, EDM2
on ImageNet) and text-conditioned ones (*e.g.*, SD3 on CC12M), without any
retraining. We will open-source the code to facilitate further research.

## 1 Introduction

Diffusion probabilistic models (DPMs) (Sohl-Dickstein et al., 2015; Ho et al., 2020; Song et al.,
2020), known simply as diffusion models, have achieved unprecedented capability improvement
of high-resolution image generation. It is well recognized that, DPMs are the most prominent
generative paradigm for a broad distribution (*i.e.*, text-to-image generation) (Podell et al., 2024;
Chen et al., 2024; Esser et al., 2024). Among DPM literature, Classifier-Free Guidance (CFG) (Ho
& Salimans, 2021) serves as an essential factor, enabling better conditional sampling. Vanilla con-
ditional sampling via DPMs introduces the conditional score function $s_t(\mathbf{x}, c) = \nabla_{\mathbf{x}_t} \log q_t(\mathbf{x}_t|c)$,
resulting in poor performance in which synthesized samples appear to be visually incoherent and not
faithful to the condition, even for large-scale models (Rombach et al., 2022). By drawing lessons
from Bayesian theory, CFG employs an interpolation between conditional and unconditional score
functions with a preset weight $\gamma$, *i.e.*,

$$s_{t,\gamma}(\mathbf{x}, c) = \gamma \nabla_{\mathbf{x}_t} \log q_t(\mathbf{x}_t|c) + (1 - \gamma)\nabla_{\mathbf{x}_t} \log q_t(\mathbf{x}_t), \tag{1}$$

in which $\nabla_{\mathbf{x}_t} \log q_t(\mathbf{x}_t)$ is the unconditional score function by annihilating the condition effect.
By doing so, DPMs turn out to formulate the underlying distribution with a gamma-powered
distribution (Bradley & Nakkiran, 2024), *i.e.*,

$$q_{t,\gamma}(\mathbf{x}|c) = q_t(\mathbf{x}|c)^\gamma q_t(\mathbf{x})^{1-\gamma}, \tag{2}$$

which is proportional to $q_t(\mathbf{x})q_t(c|\mathbf{x})^\gamma$. Enlarging $\gamma > 1$ focuses more on the classifier effect
$q_t(c|\mathbf{x})$, concentrating on better exemplars of given condition and thereby sharpening the gamma-
powered distribution. In other words, CFG is designed to promote the influence of the condition.

However, inspired by seminal works (Bradley & Nakkiran, 2024), we argue theoretically that
denoising with CFG cannot be expressed as a reciprocal of vanilla diffusion process by adding

---

[1]ReCFG, pronounced as "reconfigure", is the abbreviation for "rectified Classifier-Free Guidance".

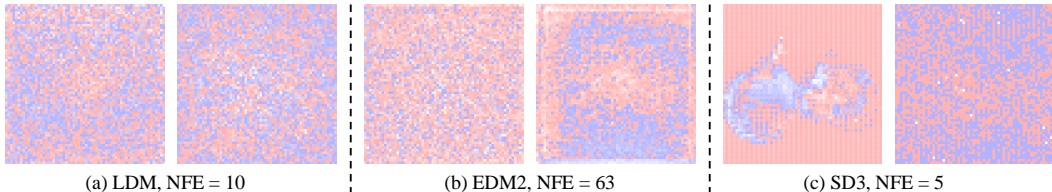

|     |     |     |
| :-: | :-: | :-: |
| (a) LDM, NFE = 10 | (b) EDM2, NFE = 63 | (c) SD3, NFE = 5 |

Figure 1: **Visualization** of the lookup table on LDM (Rombach et al., 2022), EDM2 (Karras et al., 2024b), and SD3 (Esser et al., 2024), each of which consists of the expectation ratio $\mathbb{E}_{\mathbf{x}_t}[\boldsymbol{\epsilon}_\theta(\mathbf{x}_t, c, t)]/\mathbb{E}_{\mathbf{x}_t}[\boldsymbol{\epsilon}_\theta(\mathbf{x}_t, t)]$. Each pixel represents the scale of the pixel-wise ratio, *i.e.*, color **red** implies that ratio is greater than one, while color **blue** stands for ratio smaller than one. The darker the color is, the farther the ratio appears away from one. The two images in one cell report the ratio of the first and the last denoising step under different NFEs.

Gaussian noises, since the expectation of score function of gamma-powered $q_{t,\gamma}(\mathbf{x}|c)$ is normally nonzero, violating the underlying theory of DPMs. Theoretically, score functions with zero expectation at all timesteps guarantee that the denoised $\tilde{\mathbf{x}}_0$ has expectation $\mathbb{E}[\tilde{\mathbf{x}}_0] = \frac{\alpha_0}{\alpha_T}\mathbb{E}[\mathbf{x}_T]$, thus $\mathbb{E}[\tilde{\mathbf{x}}_0] = \mathbb{E}[\mathbf{x}_0]$ and no bias on the conditional fidelity. Therefore, this theoretical flaw leaves some hidden risks during use, manifesting as a severe expectation shift phenomenon, *i.e.*, the expectation of the gamma-powered distribution will be shifted away from the ground-truth of the conditional distribution $q_t(\mathbf{x}|c)$. This is more conspicuous when applying larger $\gamma$. Fig. 2 clearly clarifies the expectation shift, in which the peak of induced distribution via CFG in red fails to coincide with that of ground-truth $q_0(\mathbf{x}_0|c)$. This theoretical flaw is known in theory (Du et al., 2023; Karras et al., 2024a; Bradley & Nakkiran, 2024), while being largely ignored in practice.

In this work, we first revisit the formulation of native CFG, theoretically confirming its flaw that we concluded above and summarizing as Theorem 1. Then, to quantitatively reveal the consequent expectation shift phenomenon by CFG, we employ a toy distribution, enjoying closed-form description of the behavior on the gamma-powered distribution. Under the toy settings, we analytically calculate the function of the precise value of expectation shift in correspondence with $\gamma$, as summarized in Theorem 2. Motivated by theoretical compatibility and canceling the expectation shift, we apply relaxation on the guidance coefficients in native CFG by circumventing the constraint that two coefficients sum to one, enabling a more flexible control on the induced distributions. To be more concrete, we propose to formulate the underlying distribution with *two* coefficients, *i.e.*,

$$q_{t,\gamma_1,\gamma_0}(\mathbf{x}|c) = q_t(\mathbf{x}|c)^{\gamma_1} q_t(\mathbf{x})^{\gamma_0}. \tag{3}$$

Aiming at consistency with the diffusion theory and thus better guidance efficacy, we specially design the constraints on $\gamma_1$ and $\gamma_0$, and theoretically confirm the feasibility. We further provide a closed-form solution to the constraints, and propose an algorithm to analytically determine $\gamma_0$ from a pre-computed lookup table in a post-hoc fashion. Thanks to the neat formulation, we can employ pixel-wise $\gamma_0$ according to the lookup table involving guidance strength $\gamma_1$, condition $c$ and timestep $t$, as demonstrated in Fig. 1. We name the above process `ReCFG`. Experiments with state-of-the-art DPMs, including both class-conditioned ones (*e.g.*, EDM2 (Karras et al., 2024b)) and text-conditioned ones (*e.g.*, SD3 (Esser et al., 2024)) under different NFEs and guidance strengths show that our `ReCFG` can achieve better guidance efficacy without retraining or extra time cost during inference stage. Hence, our work offers a new perspective on guided sampling of DPMs, encouraging more studies in the field of guided generation.

## 2 RELATED WORK

**DPMs and conditional generation.** Diffusion probabilistic model (DPM) introduces a new scheme of generative modeling, formulated by forward diffusing and reverse denoising processes (Sohl-Dickstein et al., 2015; Ho et al., 2020; Song et al., 2020). It is trained by optimizing the variational lower bound. Benefiting from this breakthrough, DPM achieves high generation fidelity, and even beat GANs on image generation. Conditional generation (Choi et al., 2021; Huang et al., 2023) takes better advantage of intrinsic intricate knowledge of data distribution, making DPM easier to

scale up and the most promising option for generative modeling. Among the literature, text-to-image generation injects the embedding of text prompts to DPM, faithfully demonstrating the text content (Podell et al., 2024; Chen et al., 2024; Esser et al., 2024).

**Classifier-Free Guidance.** Classifier-Free Guidance (CFG) serves as the successor of Classifier Guidance (CG) (Dhariwal & Nichol, 2021), circumventing the usage of a classifier for noisy images. Both CFG and CG attempt to formulate the underlying distribution by concentrating more on condition influence, achieving better conditional fidelity. Despite great success in large-scale conditional generation, CFG faces a technical flaw that the guided distribution is not theoretically guaranteed to recover the ground-truth conditional distribution (Du et al., 2023; Karras et al., 2024a; Bradley & Nakkiran, 2024). There exists a shifting issue that the expectation of guided distribution is drifted away from the correct one. This phenomenon may harm the condition faithfulness, especially for extremely broad distribution (*e.g.*, open-vocabulary synthesis).

## 3 METHOD

### 3.1 BACKGROUND ON CONDITIONAL DPMS AND CFG

Let $\mathbf{x}_0 \in \mathbb{R}^D$ be a $D$-dimensional random variable with an unknown distribution $q_0(\mathbf{x}_0|c)$, where $c \sim q(c)$ is the given condition. DPM (Sohl-Dickstein et al., 2015; Song et al., 2020; Ho et al., 2020) introduces a forward process $\{\mathbf{x}_t\}_{t \in (0,T]}$ by gradually corrupting data signal of $\mathbf{x}_0$ with Gaussian noise, *i.e.*, the following transition distribution holds for any $t \in (0, T]$:

$$q_{0t}(\mathbf{x}_t|\mathbf{x}_0, c) = q_{0t}(\mathbf{x}_t|\mathbf{x}_0) = \mathcal{N}(\alpha_t \mathbf{x}_0, \sigma_t^2 \mathbf{I}), \tag{4}$$

in which $\alpha_t, \sigma_t \in \mathbb{R}^+$ are differentiable functions of $t$ with bounded derivatives, referred to as the *noise schedule*. Let $q_t(\mathbf{x}_t|c)$ be the marginal distribution of $\mathbf{x}_t$ conditioned on $c$, DPM ensures that $q_T(\mathbf{x}_T|c) \approx \mathcal{N}(\mathbf{0}, \sigma^2 \mathbf{I})$ for some $\sigma > 0$, and the signal-to-noise-ratio (SNR) $\alpha_t^2/\sigma_t^2$ is strictly decreasing with respect to timestep $t$ (Kingma et al., 2021).

Seminal works (Kingma et al., 2021; Song et al., 2020) studied the underlying stochastic differential equation (SDE) and ordinary differential equation (ODE) theory of DPM. The forward and reverse processes are as below for any $t \in [0, T]$:

$$\mathrm{d}\mathbf{x}_t = f(t)\mathbf{x}_t \mathrm{d}t + g(t)\mathrm{d}\mathbf{w}_t, \quad \mathbf{x}_0 \sim q_0(\mathbf{x}_0|c), \tag{5}$$

$$\mathrm{d}\mathbf{x}_t = [f(t)\mathbf{x}_t - g^2(t)\nabla_{\mathbf{x}_t} \log q_t(\mathbf{x}_t|c)]\mathrm{d}t + g(t)\mathrm{d}\bar{\mathbf{w}}_t, \tag{6}$$

where $\mathbf{w}_t, \bar{\mathbf{w}}_t$ are standard Wiener processes in forward and reverse time, respectively, and $f, g$ have closed-form expressions with respect to $\alpha_t, \sigma_t$. The unknown $\nabla_{\mathbf{x}_t} \log q_t(\mathbf{x}_t|c)$ is referred to as the conditional score function. Probability flow ODE (PF-ODE) from Fokker-Planck equation enjoys the identical marginal distribution at each $t$ as that of the SDE in Eq. (6), *i.e.*,

$$\frac{\mathrm{d}\mathbf{x}_t}{\mathrm{d}t} = f(t)\mathbf{x}_t - \frac{1}{2}g^2(t)\nabla_{\mathbf{x}_t} \log q_t(\mathbf{x}_t|c). \tag{7}$$

Technically, DPM implements sampling by solving the reverse SDE or ODE from $T$ to 0. To this end, it introduces a neural network $\boldsymbol{\epsilon}_\theta(\mathbf{x}_t, c, t)$, namely the noise prediction model, to approximate the conditional score function from the given $\mathbf{x}_t$ and $c$ at timestep $t$, *i.e.*, $\boldsymbol{\epsilon}_\theta(\mathbf{x}_t, c, t) = -\sigma_t \nabla_{\mathbf{x}_t} \log q_t(\mathbf{x}_t|c)$, where the parameter $\theta$ can be optimized by the objective below:

$$\mathbb{E}_{\mathbf{x}_0, \boldsymbol{\epsilon}, c, t}[\omega_t \|\boldsymbol{\epsilon}_\theta(\mathbf{x}_t, c, t) - \boldsymbol{\epsilon}\|_2^2], \tag{8}$$

where $\omega_t$ is the weighting function, $\boldsymbol{\epsilon} \sim \mathcal{N}(\mathbf{0}, \mathbf{I})$, $c \sim q(c)$, $\mathbf{x}_t = \alpha_t \mathbf{x}_0 + \sigma_t \boldsymbol{\epsilon}$, and $t \sim \mathcal{U}[0, T]$.

For better condition fidelity, during denoising stage, CFG (Ho & Salimans, 2021) turns to use a linear interpolation between conditional and unconditional score functions, *i.e.*,

$$\nabla_{\mathbf{x}_t} \log q_{t,\gamma}(\mathbf{x}_t|c) = \gamma \nabla_{\mathbf{x}_t} \log q_t(\mathbf{x}_t|c) + (1 - \gamma)\nabla_{\mathbf{x}_t} \log q_t(\mathbf{x}_t). \tag{9}$$

Then PF-ODE can be rewrote as

$$\frac{\mathrm{d}\mathbf{x}_t}{\mathrm{d}t} = f(t)\mathbf{x}_t - \frac{1}{2}g^2(t)\nabla_{\mathbf{x}_t} \log q_{t,\gamma}(\mathbf{x}_t|c). \tag{10}$$

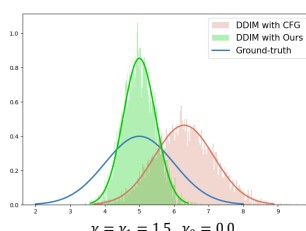 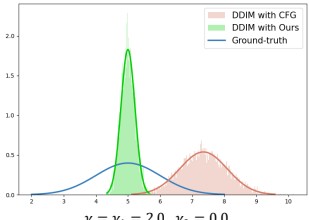 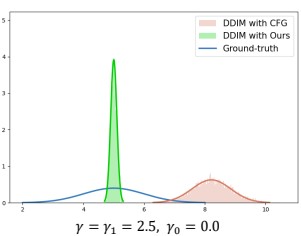

Figure 2: **Visualization** of expectation shift. The demonstrated toy data is simulated by $q_0(\mathbf{x}_0|c) \sim \mathcal{N}(c, 1)$, $q(c) \sim \mathcal{N}(0, 1)$, $q_0(\mathbf{x}_0) \sim \mathcal{N}(0, 2)$. Gamma-powered distribution $q_{0,\gamma}(\mathbf{x}_0|c)$ from CFG (Ho & Salimans, 2021) fails to recover the same conditional expectation as ground-truth due to expectation shift (*i.e.*, probability density function and histogram by DDIM (Song et al., 2021) sampler in **red**). To make a further step, larger $\gamma$ suggests more severe expectation shift, *i.e.*, the peak of $q_{0,\gamma}(\mathbf{x}_0|c)$ tends further away from $q_0(\mathbf{x}_0|c)$ (*i.e.*, probability density function in **blue**) as $\gamma$ goes from 1.5 to 2.5. As a comparison, our ReCFG successfully recovers the ground-truth expectation and smaller variance (*i.e.*, probability density function and histogram by DDIM (Song et al., 2021) sampler in **green**), consistent with the motivation of guided sampling.

We further describe the CFG under the original DDIM theory. Recall that DDIM turns out to formulate non-Markovian forward diffusing process such that the reverse denoising process obeys the distribution with parameters $\{\delta_t\}_t$ (Song et al., 2021):

$$q_\delta(\mathbf{x}_{t-1}|\mathbf{x}_t, \mathbf{x}_0, c) = q_\delta(\mathbf{x}_{t-1}|\mathbf{x}_t, \mathbf{x}_0) \sim \mathcal{N}\left(\alpha_{t-1}\mathbf{x}_0 + \sqrt{\sigma_{t-1}^2 - \delta_t^2} \cdot \frac{\mathbf{x}_t - \alpha_t\mathbf{x}_0}{\sigma_t}, \delta_t^2 \mathbf{I}\right). \quad (11)$$

Trainable generative process $p_\theta(\mathbf{x}_{t-1}|\mathbf{x}_t, c)$ is designed to leverage $q_\delta(\mathbf{x}_{t-1}|\mathbf{x}_t, \mathbf{x}_0, c)$ with a further designed denoised observation $\mathbf{f}_\theta^t$ with noise prediction model $\epsilon_\theta$, *i.e.*,

$$\mathbf{f}_\theta^t(\mathbf{x}_t, c) = \frac{1}{\alpha_t}(\mathbf{x}_t - \sigma_t \epsilon_\theta(\mathbf{x}_t, c, t)), \quad (12)$$

$$p_\theta(\mathbf{x}_{t-1}|\mathbf{x}_t, c) = \begin{cases} q_\delta(\mathbf{x}_{t-1}|\mathbf{x}_t, \mathbf{f}_\theta^t(\mathbf{x}_t, c), c), & t > 1, \\ \mathcal{N}(\mathbf{f}_\theta^t(\mathbf{x}_1), \sigma_1^2 \mathbf{I}), & t = 1. \end{cases} \quad (13)$$

DDIM proves that for any $\{\delta_t\}_t$, score matching of non-Markovian process above is equivalent to native DPM up to a constant. Under CFG setting with weight $\gamma$, we generalize the theory as below:

$$\mathbf{f}_{\theta,\gamma}^t(\mathbf{x}_t, c) = \frac{1}{\alpha_t}(\mathbf{x}_t - \sigma_t(\gamma \epsilon_\theta(\mathbf{x}_t, c, t) + (1 - \gamma)\epsilon_\theta(\mathbf{x}_t, t))), \quad (14)$$

$$p_{\theta,\gamma}(\mathbf{x}_{t-1}|\mathbf{x}_t, c) = \begin{cases} q_\delta(\mathbf{x}_{t-1}|\mathbf{x}_t, \mathbf{f}_{\theta,\gamma}^t(\mathbf{x}_t, c), c), & t > 1, \\ \mathcal{N}(\mathbf{f}_{\theta,\gamma}^t(\mathbf{x}_1, c), \sigma_1^2 \mathbf{I}), & t = 1. \end{cases} \quad (15)$$

Native DDIM theory still holds since $q_\delta(\mathbf{x}_{t-1}|\mathbf{x}_t, \mathbf{x}_0, c) = q_\delta(\mathbf{x}_{t-1}|\mathbf{x}_t, \mathbf{x}_0)$, *i.e.*, with the definition

$$J_{\delta,\gamma}(\epsilon_\theta) = \mathbb{E}_{q_\delta(\mathbf{x}_{0:T}|c)}[\log q_\delta(\mathbf{x}_{1:T}|\mathbf{x}_0, c) - \log p_{\theta,\gamma}(\mathbf{x}_{0:T}|c)], \quad (16)$$

we have the following theorem. Proof is in Appendix A.1.

**Theorem 1.** *For any $\{\delta_t\}_t$ and $\gamma > 1$, $J_{\delta,\gamma}$ is equivalent to native DPM under CFG up to a constant. However, denoising with CFG is not a reciprocal of the original diffusion process with Gaussian noise due to nonzero expectation of unconditional score function $\mathbb{E}_{q_t(\mathbf{x}_t|c)}[\nabla_{\mathbf{x}_t} \log q_t(\mathbf{x}_t)]$.*

**Remark 1.** *$\epsilon_\theta(\mathbf{x}_t, c, t)$ and $\epsilon_\theta(\mathbf{x}_t, t)$ are proportional to $\nabla_{\mathbf{x}_t} \log q_t(\mathbf{x}_t|c)$ and $\nabla_{\mathbf{x}_t} \log q_t(\mathbf{x}_t)$ with coefficients being each minus standard deviation respectively, and empirically we use the same fixed variance for both $q(\mathbf{x}_{t-1}|\mathbf{x}_t, \mathbf{x}_0, c)$ and $q(\mathbf{x}_{t-1}|\mathbf{x}_t, \mathbf{x}_0)$. Therefore, Theorem 1 is consistent with the original CFG using score functions in Eq. (9).*

### 3.2 MISCONCEPTIONS ABOUT CFG ON EXPECTATION SHIFT

CFG is designed to concentrate on better exemplars for each denoising step by sharpening the gamma-powered distribution as below (Bradley & Nakkiran, 2024)

$$q_{t,\gamma}(\mathbf{x}|c) = q_t(\mathbf{x}|c)^\gamma q_t(\mathbf{x})^{1-\gamma}. \quad (17)$$

We first generalize the counterexample in Bradley & Nakkiran (2024) to confirm the expectation shift phenomenon. For VE-SDE with deterministic sampling recipe, we consider the 1-dimensional distribution with $q_0(\mathbf{x}_0|c) \sim \mathcal{N}(c, 1)$, $q(c) \sim \mathcal{N}(0, 1)$, $q_0(\mathbf{x}_0) \sim \mathcal{N}(0, 2)$. Then we can formulate the forward process and score functions as below:

$$q_t(\mathbf{x}_t|c) \sim \mathcal{N}(c, 1+t), \tag{18}$$

$$q_t(\mathbf{x}_t) \sim \mathcal{N}(0, 2+t), \tag{19}$$

$$\nabla_{\mathbf{x}_t} \log q_t(\mathbf{x}_t|c) = -\frac{\mathbf{x}_t - c}{1+t}, \tag{20}$$

$$\nabla_{\mathbf{x}_t} \log q_t(\mathbf{x}_t) = -\frac{\mathbf{x}_t}{2+t}. \tag{21}$$

We state the theorem below describing the expectation shift. Proof is addressed in Appendix A.2.

**Theorem 2.** *Denote by $q_{0,\gamma}^{\text{deter}}(\mathbf{x}_0|c)$ the conditional distribution by solving PF-ODE in Eq. (10) with CFG weight $\gamma > 1$. Then $q_{0,\gamma}^{\text{deter}}(\mathbf{x}_0|c)$ follows the closed-form expression as below.*

$$q_{0,\gamma}^{\text{deter}}(\mathbf{x}_0|c) \sim \mathcal{N}\left(c\phi(\gamma, T), 2^{1-\gamma}\frac{T+1}{(T+1)^\gamma(T+2)^{1-\gamma}}\right), \tag{22}$$

*in which*

$$\phi(\gamma, T) = 2^{\frac{1-\gamma}{2}}\left(\frac{1}{(T+1)^{\frac{\gamma}{2}}(T+2)^{\frac{1-\gamma}{2}}} + \frac{\gamma}{2}\int_0^T (s+1)^{-\frac{\gamma+2}{2}}(s+2)^{-\frac{1-\gamma}{2}}\mathrm{d}s\right). \tag{23}$$

*Specifically, when $T \to +\infty$, denote by $\phi(\gamma)$ with*

$$\phi(\gamma) = \lim_{T \to +\infty} \phi(\gamma, T), \tag{24}$$

*we have $\phi(\gamma) \geqslant \gamma\frac{7}{15}\left(\frac{10}{7}\right)^{\frac{5-\gamma}{2}}$ for $\gamma \in [1, 3]$, $\phi(1) = 1$, $\phi(3) = 2$, $\phi(\gamma) \geqslant 2$ for all $\gamma > 3$, and*

$$q_{0,\gamma}^{\text{deter}}(\mathbf{x}_0|c) \sim \mathcal{N}(c\phi(\gamma), 2^{1-\gamma}). \tag{25}$$

*Furthermore, we have closed-form expression for $\phi(\gamma)$ when $\gamma \in \mathbb{N}$ and $\gamma > 1$, i.e.,*

$$\phi(\gamma) = \begin{cases} 2^{-n}\left(\displaystyle\sum_{k=0}^n C_n^k \frac{2n+1}{2n-2k+1}\right), & \gamma = 2n+1, \\[4mm] 2^{-\frac{1}{2}}\left(\sqrt{2} - \frac{1}{2}\log\frac{\sqrt{2}-1}{\sqrt{2}+1}\right), & \gamma = 2, \\[4mm] \dfrac{(2n-1)!!\sqrt{2}n}{(2n)!!2^n}\left(\left(\displaystyle\sum_{k=2}^n \frac{1}{k}\frac{(2k)!!2^{k-\frac{1}{2}}}{(2k-1)!!}\right) + 2\sqrt{2} - \log\frac{\sqrt{2}-1}{\sqrt{2}+1}\right), & \gamma = 2n \geqslant 4. \end{cases} \tag{26}$$

However, note that the ground-truth conditional distribution $q_0(\mathbf{x}_0|c) \sim \mathcal{N}(c, 1)$, indicating that the ground-truth expectation is equal to $c$. That is to say, denoising with CFG achieves at least twice as large expectation as the ground-truth one. Fig. 2 clearly describes the phenomenon.

## 3.3 RECTIFIED CLASSIFIER-FREE GUIDANCE

Recall that the constraint of the two coefficients with summation one disables the compatibility with diffusion theory and indicates expectation shift. Theorem 2 quantitatively describes the expectation shift, claiming that the two coefficients of conditional and unconditional score functions in Eq. (9) dominate both the expectation and variance of $q_{0,\gamma}^{\text{deter}}(\mathbf{x}_0|c)$. To this end, we propose to rectify CFG with relaxation on the guidance coefficients, *i.e.*,

$$\nabla_{\mathbf{x}_t} \log q_{t,\gamma_1,\gamma_0}(\mathbf{x}_t|c) = \gamma_1 \otimes \nabla_{\mathbf{x}_t} \log q_t(\mathbf{x}_t|c) + \gamma_0 \otimes \nabla_{\mathbf{x}_t} \log q_t(\mathbf{x}_t), \tag{27}$$

in which $\gamma_1$, $\gamma_0 \in \mathbb{R}^D$ are functions with respect to condition $c$ and timestep $t$, and $\otimes$ indicates element-wise product. Denote by $q_{0,\gamma_1,\gamma_0}(\mathbf{x}_0|c)$ the attached conditional distribution following PF-ODE in Eq. (10) with $\nabla_{\mathbf{x}_t} \log q_{t,\gamma_1,\gamma_0}(\mathbf{x}_t|c)$.

To make guided sampling compatible with the diffusion theory and annihilate expectation shift, it suffices to choose more appropriate $\gamma_1$ and $\gamma_0$ according to input condition $c$ and timestep $t$. Intuitively, we need the constraint such that:

- Each component $\gamma_{1,i} > 1$ for strengthened conditional fidelity,

- Denoising with PF-ODE and Eq. (27) is theoretically the reciprocal of forward process, thus $q_{0,\gamma_1,\gamma_0}(\mathbf{x}_0|c)$ enjoys the same expectation as ground-truth $q_0(\mathbf{x}_0|c)$,

- $q_{0,\gamma_1,\gamma_0}(\mathbf{x}_0|c)$ enjoys smaller or the same variance as ground-truth $q_0(\mathbf{x}_0|c)$ for sharper distribution and better exemplars.

In the sequel, we omit $\otimes$ for simplicity. We first focus on the compatibility with the diffusion theory. We have claimed in Theorem 1 that CFG cannot satisfy the diffusion theory due to nonzero $\mathbb{E}_{q_t(\mathbf{x}_t|c)}[\nabla_{\mathbf{x}_t} \log q_t(\mathbf{x}_t)]$. To this end, it suffices to annihilate the expectation shift as below:

$$\mathbb{E}_{q_t(\mathbf{x}_t|c)}[\nabla_{\mathbf{x}_t} \log q_{t,\gamma_1,\gamma_0}(\mathbf{x}_t|c)] = -\frac{1}{\sigma_t}\mathbb{E}_{q_t(\mathbf{x}_t|c)}[\gamma_1\boldsymbol{\epsilon}_\theta(\mathbf{x}_t,c,t) + \gamma_0\boldsymbol{\epsilon}_\theta(\mathbf{x}_t,t)] = 0. \quad (28)$$

To confirm the feasibility and precisely describe the expectation of $q_{0,\gamma_1,\gamma_0}(\mathbf{x}_0|c)$, resembling Eqs. (14) and (15) we can write:

$$\mathbf{f}_{\theta,\gamma_1,\gamma_0}^t(\mathbf{x}_t,c) = \frac{1}{\alpha_t}(\mathbf{x}_t - \sigma_t(\gamma_1\boldsymbol{\epsilon}_\theta(\mathbf{x}_t,c,t) + \gamma_0\boldsymbol{\epsilon}_\theta(\mathbf{x}_t,t))), \quad (29)$$

$$p_{\theta,\gamma_1,\gamma_0}(\mathbf{x}_{t-1}|\mathbf{x}_t,c) = \begin{cases} q_\delta(\mathbf{x}_{t-1}|\mathbf{x}_t, \mathbf{f}_{\theta,\gamma_1,\gamma_0}^t(\mathbf{x}_t,c),c), & t > 1, \\ \mathcal{N}(\mathbf{f}_{\theta,\gamma_1,\gamma_0}^t(\mathbf{x}_1,c), \sigma_1^2\mathbf{I}), & t = 1. \end{cases} \quad (30)$$

And we have the following theorem, in which proof is addressed in Appendix A.3.

**Theorem 3.** *Let* $\mathbf{x}_t \sim q_t(\mathbf{x}_t|c)$, $\tilde{\mathbf{x}}_t \sim p_{\theta,\gamma_1,\gamma_0}(\tilde{\mathbf{x}}_t|c)$ *induced from DDIM sampler in Eq. (30). Assume that all* $\delta_t = 0$, *denote by* $\Delta_t$ *the difference between expectation of* $\mathbf{x}_t$ *and* $\tilde{\mathbf{x}}_t$, *i.e.,*

$$\Delta_t = \mathbb{E}_{q_t(\mathbf{x}_t|c)}[\mathbf{x}_t] - \mathbb{E}_{p_{\theta,\gamma_1,\gamma_0}(\tilde{\mathbf{x}}_t|c)}[\tilde{\mathbf{x}}_t]. \quad (31)$$

*Then we have the following recursive equality:*

$$\Delta_{t-1} = \frac{\sigma_{t-1}}{\sigma_t}\Delta_t - (\sigma_{t-1} - \frac{\alpha_{t-1}}{\alpha_t}\sigma_t)\mathbb{E}_{\tilde{\mathbf{x}}_t}[(\gamma_1 - 1)\boldsymbol{\epsilon}_\theta(\tilde{\mathbf{x}}_t,c,t) + \gamma_0\boldsymbol{\epsilon}_\theta(\tilde{\mathbf{x}}_t,t)]). \quad (32)$$

*Specifically, when* $\Delta_t = 0$, *we have:*

$$\Delta_{t-1} = -(\sigma_{t-1} - \frac{\alpha_{t-1}}{\alpha_t}\sigma_t)\mathbb{E}_{\mathbf{x}_t}[(\gamma_1 - 1)\boldsymbol{\epsilon}_\theta(\mathbf{x}_t,c,t) + \gamma_0\boldsymbol{\epsilon}_\theta(\mathbf{x}_t,t)]). \quad (33)$$

Theorem 3 studies the difference between expectation of denoising with Eq. (27) and the ground-truth. Note that $\mathbb{E}_{\mathbf{x}_t}[\boldsymbol{\epsilon}_\theta(\mathbf{x}_t,c,t)] = \mathbb{E}_{\mathbf{x}_t}[\mathbb{E}_{q_t(\boldsymbol{\epsilon}|\mathbf{x}_t)}[\boldsymbol{\epsilon}|\mathbf{x}_t]] = \mathbb{E}_{\mathbf{x}_t}[\boldsymbol{\epsilon}] = 0$, we have

$$\mathbb{E}_{\mathbf{x}_t}[(\gamma_1 - 1)\boldsymbol{\epsilon}_\theta(\mathbf{x}_t,c,t) + \gamma_0\boldsymbol{\epsilon}_\theta(\mathbf{x}_t,t)] = \mathbb{E}_{\mathbf{x}_t}[\gamma_1\boldsymbol{\epsilon}_\theta(\mathbf{x}_t,c,t) + \gamma_0\boldsymbol{\epsilon}_\theta(\mathbf{x}_t,t)], \quad (34)$$

which coincides with Eq. (28), indicating the feasibility and a closed-form solution given $c$ and $t$.

As for variance, however, normally we cannot calculate the variance of $p_{\theta,\gamma_1,\gamma_0}(\mathbf{x}_t|c)$. Instead, we study the variance in Sec. 3.2 as an empirical evidence, where proof is in Appendix A.4.

**Theorem 4.** *Under settings in Theorem 2, denote by* $q_{0,\gamma_1,\gamma_0}^{\mathrm{deter}}(\mathbf{x}_0|c)$ *the conditional distribution by PF-ODE and deterministic sampling with* $\gamma_1$ *and* $\gamma_0$ *as in Eq. (27). Then we have*

$$\mathrm{var}_{q_{0,\gamma_1,\gamma_0}^{\mathrm{deter}}(\mathbf{x}_0|c)}[\mathbf{x}_0] = 2^{\gamma_0}(T+1)^{1-\gamma_1}(T+2)^{-\gamma_0}. \quad (35)$$

According to Theorem 4, variance of $q_{0,\gamma_1,\gamma_0}^{\mathrm{deter}}(\mathbf{x}_0|c)$ is guaranteed to be smaller than the ground-truth $\mathrm{var}_{q_0(\mathbf{x}_0|c)}[\mathbf{x}_0] = 1$ when each $\gamma_{0,i} \leqslant 0$ and $\gamma_{1,i} + \gamma_{0,i} \geqslant 1$, especially when $T \to +\infty$.

Now we formally propose the constraints. First, we need $\gamma_{1,i} > 1$ for strengthened conditional fidelity. Then for expectation, it is noteworthy that $\Delta_T = 0$ satisfies the assumption in Theorem 3. Therefore by induction, it is feasible to annihilate $\Delta_0$ by annihilation of Eq. (28). Finally as for variance, we empirically set $\gamma_{0,i} \leqslant 0$ and $\gamma_{1,i} + \gamma_{0,i} \geqslant 0$.

Practically, we can determine $\gamma_0$ according to the guidance strength $\gamma_1$, condition $c$, and timestep $t$, according to the closed-form solution of Eq. (33). Concretely, given condition $c$, it is feasible to pre-compute a collection of $\{(\epsilon_\theta(\mathbf{x}_t, c, t), \epsilon_\theta(\mathbf{x}_t, t))\}_t$ by traversing $q_0(\mathbf{x}_0|c)$, and maintain a lookup table consisting of $\mathbb{E}_{\mathbf{x}_t}[\epsilon_\theta(\mathbf{x}_t, c, t)]/\mathbb{E}_{\mathbf{x}_t}[\epsilon_\theta(\mathbf{x}_t, t)]$. Then given any $\gamma_1$, we can directly achieve $\gamma_0$ by multiplying $-(\gamma_1 - 1)$ with the expectation ratio. Pseudo-code is addressed in Appendix B.

We make further discussion about `ReCFG`. By Cauchy-Schwarz inequality and Eq. (33) we have:

$$\|\Delta_{t-1}\|_2^2 \leqslant (\sigma_{t-1} - \frac{\alpha_{t-1}}{\alpha_t}\sigma_t)^2 \mathbb{E}_{\mathbf{x}_t}[\|(\gamma_1 - 1)\epsilon_\theta(\mathbf{x}_t, c, t) + \gamma_0\epsilon_\theta(\mathbf{x}_t, t)\|_2^2]. \tag{36}$$

Then we can define the objective resembling DPMs as below, optimizing reversely from $t = T$ to 0.

$$\mathcal{L}_{\gamma_1, \gamma_0} = \mathbb{E}_{\mathbf{x}_t, t}[\|(\gamma_1 - 1)\epsilon_\theta(\mathbf{x}_t, c, t) + \gamma_0\epsilon_\theta(\mathbf{x}_t, t)\|_2^2]. \tag{37}$$

Resembling Theorem 1, with Eq. (37), we can also show the compatibility of `ReCFG` with DDIM, which is summarized as the theorem below. Proof is addressed in Appendix A.5

**Theorem 5.** *For any $\{\delta_t\}_t$, `ReCFG` with $\mathcal{L}_{\gamma_1, \gamma_0}$ is compatible with native DPM up to a constant.*

## 4 EXPERIMENTS

### 4.1 EXPERIMENTAL SETUPS

**Datasets and baselines.** We apply `ReCFG` to previous seminal DPMs, including LDM (Rombach et al., 2022) on ImageNet 256 (Deng et al., 2009), EDM2 (Karras et al., 2024b) on ImageNet 512, and SD3 (Esser et al., 2024) on CC12M (Changpinyo et al., 2021), respectively.

**Evaluation metrics.** As for LDM and EDM2, we draw 50,000 samples for Fréchet Inception Distance (FID) (Heusel et al., 2017) to evaluate the fidelity of the synthesized images. We further use Improved Precision (Prec.) and Recall (Rec.) (Kynkäänniemi et al., 2019) to separately measure sample fidelity (Precision) and diversity (Recall). As for SD3, following the official implementation, we use CLIP Score (CLIP-S) (Radford et al., 2021; Hessel et al., 2021) and FID on CLIP features (Sauer et al., 2021) on 1,000 samples to evaluate conditional faithfulness and fidelity of the synthesized images, respectively. Both two metrics are evaluated on the MS-COCO validation split (Lin et al., 2015).

**Implementation details.** We implement `ReCFG` with NVIDIA A100 GPUs, and use pre-trained LDM[2], EDM2[3], and SD3[4] provided in official implementation. We reproduce all the experiments with official and more other configurations including NFEs and guidance strengths.

### 4.2 RESULTS ON TOY EXAMPLE IN SECTION 3.2

We first confirm the effectiveness of our method on toy data, as presented in Sec. 3.2. Given the closed-form expressions of score functions, we are able to precisely describe the distribution of both gamma-powered distribution $q_{0,\gamma}(\mathbf{x}_0|c)$ by native CFG and $q_{0,\gamma_1,\gamma_0}(\mathbf{x}_0|c)$ by `ReCFG`. The theoretical and numerical DDIM-based simulation value of probability density functions of both $q_{0,\gamma}(\mathbf{x}_0|c)$ and $q_{0,\gamma_1,\gamma_0}(\mathbf{x}_0|c)$ are shown in Fig. 2. It is noteworthy that native CFG drifts the expectation of $q_{0,\gamma}(\mathbf{x}_0|c)$ further away from the peak of the ground-truth $q_0(\mathbf{x}_0|c)$ as $\gamma$ becomes larger, consistent with Theorem 2. As a comparison, the peaks of $q_{0,\gamma_1,\gamma_0}(\mathbf{x}_0|c)$ and $q_0(\mathbf{x}_0|c)$ coincide, while $q_{0,\gamma_1,\gamma_0}(\mathbf{x}_0|c)$ is sharpened with smaller variance. Therefore, by adopting relaxation on coefficients $\gamma_1$ and $\gamma_0$ with specially proposed constraints, our `ReCFG` manages to annihilate expectation shift, enabling better guidance and thus better conditional fidelity.

---

[2]https://github.com/CompVis/latent-diffusion
[3]https://github.com/NVlabs/edm2
[4]https://huggingface.co/stabilityai/stable-diffusion-3-medium-diffusers

Table 1: **Sample quality** on ImageNet (Deng et al., 2009). For clearer demonstration, settings of $\gamma_1 + \gamma_0 = 1$ (*i.e.* falling into native CFG) are highlighted in **gray**.

Table 2: **Sample quality** on CC12M (Chang-pinyo et al., 2021) For clearer demonstration, settings of $\gamma_1 + \gamma_0 = 1$ (*i.e.* falling into native CFG) are highlighted in **gray**.

| ImageNet 256x256, LDM (Rombach et al., 2022) | | | | | |
| --- | --- | --- | --- | --- | --- |
| $\gamma_1$ | $\gamma_0$ | NFE ($\downarrow$) | FID ($\downarrow$) | Prec. ($\uparrow$) | Rec. ($\uparrow$) |
| 5.0 | -4.0 | 20 | 18.87 | **0.95** | 0.15 |
| 5.0 | ReCFG | 20 | **16.95** | **0.91** | **0.18** |
| 3.0 | -2.0 | 20 | 11.46 | **0.94** | 0.27 |
| 3.0 | ReCFG | 20 | **9.78** | 0.91 | **0.32** |
| 5.0 | -4.0 | 10 | 16.78 | **0.94** | 0.16 |
| 5.0 | ReCFG | 10 | **14.46** | 0.89 | **0.22** |
| 3.0 | -2.0 | 10 | 10.13 | 0.91 | 0.28 |
| 3.0 | ReCFG | 10 | **8.26** | 0.91 | **0.33** |

| ImageNet 512x512, EDM2-S (Karras et al., 2024b) | | | | | |
| --- | --- | --- | --- | --- | --- |
| $\gamma_1$ | $\gamma_0$ | NFE ($\downarrow$) | FID ($\downarrow$) | Prec. ($\uparrow$) | Rec. ($\uparrow$) |
| 3.0 | -2.0 | 63 | 6.81 | **0.85** | 0.43 |
| 3.0 | ReCFG | 63 | **5.59** | 0.84 | 0.43 |
| 2.5 | -1.5 | 63 | 5.87 | **0.85** | 0.46 |
| 2.5 | ReCFG | 63 | **4.84** | 0.84 | **0.48** |
| 2.0 | -1.0 | 63 | 4.18 | **0.85** | 0.52 |
| 2.0 | ReCFG | 63 | **3.61** | 0.84 | 0.52 |
| 1.4 | -0.4 | 63 | 2.29 | 0.83 | 0.59 |
| 1.4 | ReCFG | 63 | **2.23** | 0.83 | 0.59 |

| CC12M 512x512, SD3 (Esser et al., 2024) | | | | |
| --- | --- | --- | --- | --- |
| $\gamma_1$ | $\gamma_0$ | NFE ($\downarrow$) | CLIP-S ($\uparrow$) | FID ($\downarrow$) |
| 7.5 | -6.5 | 25 | 0.268 | 72.24 |
| 7.5 | ReCFG | 25 | **0.270** | **71.83** |
| 5.0 | -4.0 | 25 | 0.267 | 72.37 |
| 5.0 | ReCFG | 25 | **0.268** | **71.95** |
| 2.5 | -1.5 | 25 | 0.262 | 70.50 |
| 2.5 | ReCFG | 25 | **0.263** | **69.99** |
| 7.5 | -6.5 | 10 | 0.262 | 82.71 |
| 7.5 | ReCFG | 10 | **0.263** | **76.05** |
| 5.0 | -4.0 | 10 | 0.268 | 72.55 |
| 5.0 | ReCFG | 10 | **0.269** | **70.31** |
| 2.5 | -1.5 | 10 | 0.265 | 71.17 |
| 2.5 | ReCFG | 10 | 0.265 | **68.68** |
| 7.5 | -6.5 | 5 | 0.209 | 156.60 |
| 7.5 | ReCFG | 5 | **0.229** | **140.89** |
| 5.0 | -4.0 | 5 | 0.248 | 115.51 |
| 5.0 | ReCFG | 5 | **0.258** | **101.82** |
| 2.5 | -1.5 | 5 | 0.261 | 101.95 |
| 2.5 | ReCFG | 5 | **0.263** | **96.80** |

Table 3: **Variance** of lookup table over condition $c$. Note that we employ pixel-wise lookup table involving timestep $t$. We report the the mean and variance of lookup table over $c$, which is computed by averaging on all timesteps $t$ and pixels.

| Config. | LDM, NFE $= 10$ | EDM2, NFE $= 63$ | SD3, NFE $= 5$ | SD3, NFE $= 10$ |
| --- | --- | --- | --- | --- |
| Expectation Ratio | $1.0050 \pm 0.0012$ | $1.0060 \pm 0.0119$ | $1.0250 \pm 0.0369$ | $1.0125 \pm 0.0281$ |

## 4.3 RESULTS ON REAL DATASETS

We conduct extensive experiments on state-of-the-art DPMs to quantitatively convey the efficacy of ReCFG. Results are reported in Tabs. 1 and 2. We can tell that ReCFG is capable of better performance on both class-conditioned and text-conditioned DPMs under various guidance strengths and NFEs. Additionally, ReCFG achieves better CLIP-S especially on small NFEs and large guidance strength, indicating better conditional fidelity on open-vocabulary synthesis.

## 4.4 ANALYSES

**Variance of lookup table over condition $c$.** Note that we need to pre-compute the lookup table consisting of expectation ratios for all conditions $c$, which is time-consuming and impractical for open-vocabulary distributions (*e.g.*, text-conditioned DPMs). In Tab. 3 we report the mean and variance of expectation ratios over condition $c$, which is averaged on all timesteps and pixels. It is noteworthy that the variance of text-conditioned DPMs is larger than that of class-conditioned ones, while both of which is insignificant compared to the mean. Therefore, it is feasible to prepare the lookup table for only part of all potential conditions and use the mean for all conditions, serving as a practical strategy to improve time efficiency.

**Ablation studies.** Recall that we pre-compute the lookup table by traversing the dataset $q_0(\mathbf{x}_0|c)$. We conduct comprehensive ablation studies to convey a direct and clear picture of the efficacy of ReCFG under different numbers of traversals, as reported in Tabs. 4 and 5. We can conclude that larger number of traversals suggests better guidance performance, yet improvements from 100 to 500 traversals are relatively inconspicuous. In other words, employment of 500 samples per condition is adequate to serve as an empirical setting.

Table 4: **Ablation study** of the number of traversals (the number after `ReCFG`) for lookup table on ImageNet (Deng et al., 2009). For clearer demonstration, baselines of native CFG are highlighted in **gray**.

| ImageNet 256x256, LDM (Rombach et al., 2022) | | | | | |
|---|---|---|---|---|---|
| $\gamma_1$ | $\gamma_0$ | NFE ($\downarrow$) | FID ($\downarrow$) | Prec. ($\uparrow$) | Rec. ($\uparrow$) |
| 3.0 | -2.0 | 10 | 10.13 | 0.91 | 0.28 |
| 3.0 | `ReCFG-10` | 10 | 8.88 | **0.92** | 0.30 |
| 3.0 | `ReCFG-100` | 10 | 8.70 | **0.92** | 0.31 |
| 3.0 | `ReCFG-500` | 10 | **8.26** | 0.91 | **0.33** |
| ImageNet 512x512, EDM2-S (Karras et al., 2024b) | | | | | |
| $\gamma_1$ | $\gamma_0$ | NFE ($\downarrow$) | FID ($\downarrow$) | Prec. ($\uparrow$) | Rec. ($\uparrow$) |
| 2.5 | -1.5 | 63 | 5.87 | **0.85** | 0.46 |
| 2.5 | `ReCFG-10` | 63 | 5.06 | 0.84 | 0.47 |
| 2.5 | `ReCFG-100` | 63 | 4.99 | 0.84 | 0.45 |
| 2.5 | `ReCFG-500` | 63 | **4.84** | 0.84 | **0.48** |
| 2.0 | -1.0 | 63 | 4.18 | **0.85** | 0.52 |
| 2.0 | `ReCFG-10` | 63 | 3.70 | 0.84 | 0.52 |
| 2.0 | `ReCFG-100` | 63 | 3.66 | 0.84 | 0.52 |
| 2.0 | `ReCFG-500` | 63 | **3.61** | 0.84 | 0.52 |

Table 5: **Ablation study** of the number of traversals (the number after `ReCFG`) for lookup table on CC12M (Changpinyo et al., 2021). For clearer demonstration, baselines of native CFG are highlighted in **gray**.

| CC12M 512x512, SD3 (Esser et al., 2024) | | | | |
|---|---|---|---|---|
| $\gamma_1$ | $\gamma_0$ | NFE ($\downarrow$) | CLIP-S ($\uparrow$) | FID ($\downarrow$) |
| 5.0 | -4.0 | 25 | 0.267 | 72.37 |
| 5.0 | `ReCFG-10` | 25 | 0.267 | 72.15 |
| 5.0 | `ReCFG-100` | 25 | **0.268** | 72.03 |
| 5.0 | `ReCFG-500` | 25 | **0.268** | **71.95** |
| 5.0 | -4.0 | 10 | 0.268 | 72.55 |
| 5.0 | `ReCFG-10` | 10 | 0.268 | 71.61 |
| 5.0 | `ReCFG-100` | 10 | 0.268 | 70.64 |
| 5.0 | `ReCFG-500` | 10 | **0.269** | **70.31** |
| 5.0 | -4.0 | 5 | 0.248 | 115.51 |
| 5.0 | `ReCFG-10` | 5 | 0.252 | 107.09 |
| 5.0 | `ReCFG-100` | 5 | 0.256 | 103.25 |
| 5.0 | `ReCFG-500` | 5 | **0.258** | **101.82** |

**Pixel-wise lookup table.** `ReCFG` enables pixel-specific guidance coefficients $\gamma_1$ and $\gamma_0$, thanks to the closed-form solution to Eq. (33), *i.e.*, we can assign $\gamma_0$ for each pixel by maintaining the lookup table of pixel-wise expectation ratios. Fig. 1 demonstrates the ratios on LDM, EDM2, and SD3 under different NFEs. It is noteworthy that there appears no general rules on the relation between $\gamma_1$ and $\gamma_0$, indicating that trivially setting $\gamma_1$ and $\gamma_0$ to be scalars is less reasonable. As a comparison, our method makes it possible to employ more precise control on guided sampling in a simple and post-hoc fashion without further fine-tuning, enabling better performance.

## 4.5 DISCUSSIONS

Classifier-Free Guidance is designed from Bayesian theory to facilitate conditional sampling, yet appears incompatible with original diffusion theory. Therefore, we believe `ReCFG` is attached to great importance on guided sampling by fixing the theoretical flaw of CFG. Despite the success on better conditional fidelity, our algorithm has several potential limitations. We need to pre-compute the lookup table by traversing the dataset to achieve rectified coefficients for each condition. Although we conduct extensive ablation studies on the number of traversals and variance over condition $c$, providing an adequate strategy especially for open-vocabulary datasets on text-conditioned synthesis, the optimal strategy is unexplored. Besides, we at present cannot provide precise control on variance of `ReCFG`, and turn to employ empirical values. Therefore, how to further conquer these problems (*e.g.*, employing a predictor network $\boldsymbol{\omega}(c, t)$ for better $\gamma_0$ on open-vocabulary datasets according to Eq. (37)) will be an interesting avenue for future research. Although leaving the variance behavior unexplored, we hope that `ReCFG` will encourage the community to close the gap in the future.

## 5 CONCLUSION

In this paper, we analyze the theoretical flaws of native Classifier-Free Guidance technique and the induced expectation shift phenomenon. We theoretically claim the exact value of expectation shift on a toy distribution. Introducing a relaxation on coefficients of CFG and novel constraints, we manage to complete the theory of guided sampling by fixing the incompatibility between CFG and diffusion theory. Accordingly, thanks to the closed-form solution to the constraints, we propose `ReCFG`, a post-hoc algorithm aiming at more faithful guided sampling by determining the coefficients from a pre-computed lookup table. We further study the behavior of the lookup table, proposing an adequate strategy for better time efficiency in practice. Comprehensive experiments demonstrate the efficacy of our method on various state-of-the-art DPMs under different NFEs and guidance strengths.

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

APPENDIX

## A  PROOFS AND DERIVATIONS

In this section, we will prove the theorems stated in the main manuscript.

### A.1  PROOF OF THEOREM 1

We first claim two lemmas which are crucial for the proof.

**Lemma 1.** *Let $g(\mathbf{x}_t)$ and $h(\mathbf{x}_t, \boldsymbol{\epsilon})$ be integrable functions, then the following equality holds.*

$$\mathbb{E}_{q(\mathbf{x})}[\langle g(\mathbf{x}), \mathbb{E}_{q(\boldsymbol{\epsilon}|\mathbf{x})}[h(\mathbf{x}, \boldsymbol{\epsilon})|\mathbf{x}]\rangle] = \mathbb{E}_{q(\mathbf{x},\boldsymbol{\epsilon})}[\langle g(\mathbf{x}), h(\mathbf{x}, \boldsymbol{\epsilon})\rangle], \tag{S1}$$

*in which $\langle \cdot, \cdot \rangle$ is inner product.*

*Proof of Lemma 1.* Note that

$$\mathbb{E}_{q(\mathbf{x})}[\langle g(\mathbf{x}), \mathbb{E}_{q(\boldsymbol{\epsilon}|\mathbf{x})}[h(\mathbf{x}, \boldsymbol{\epsilon})|\mathbf{x}]\rangle] = \int \langle g(\mathbf{x}), \mathbb{E}_{q(\boldsymbol{\epsilon}|\mathbf{x})}[h(\mathbf{x}, \boldsymbol{\epsilon})|\mathbf{x}]\rangle q(\mathbf{x})\mathrm{d}\mathbf{x} \tag{S2}$$

$$= \int \langle g(\mathbf{x}), \int h(\mathbf{x}, \boldsymbol{\epsilon}) q(\boldsymbol{\epsilon}|\mathbf{x})\mathrm{d}\boldsymbol{\epsilon}\rangle q(\mathbf{x})\mathrm{d}\mathbf{x} \tag{S3}$$

$$= \iint \langle g(\mathbf{x}), h(\mathbf{x}, \boldsymbol{\epsilon})\rangle q(\mathbf{x})q(\boldsymbol{\epsilon}|\mathbf{x})\mathrm{d}\boldsymbol{\epsilon}\mathrm{d}\mathbf{x} \tag{S4}$$

$$= \mathbb{E}_{q(\mathbf{x},\boldsymbol{\epsilon})}[\langle g(\mathbf{x}), h(\mathbf{x}, \boldsymbol{\epsilon})\rangle], \tag{S5}$$

in which Eq. (S4) is by linearity of integral. □

**Lemma 2.** *The following equality of expectation holds:*

$$\mathbb{E}_{\mathbf{x}}[\boldsymbol{\epsilon}_\theta(\mathbf{x}, t)] = \frac{1}{\sigma_t}\mathbb{E}_{\mathbf{x}}[\mathbf{x}] - \frac{\alpha_t}{\sigma_t}\mathbb{E}_{c,\mathbf{x}_0,\mathbf{x}}[\mathbf{x}_0]. \tag{S6}$$

*Proof of Lemma 2.* Note that

$$\nabla_{\mathbf{x}} \log q_t(\mathbf{x}) = \frac{\nabla_{\mathbf{x}} q_t(\mathbf{x})}{q_t(\mathbf{x})} \tag{S7}$$

$$= \frac{\nabla_{\mathbf{x}} \int q_t(\mathbf{x}|c)q(c)\mathrm{d}c}{q_t(\mathbf{x})} \tag{S8}$$

$$= \frac{\int \nabla_{\mathbf{x}} q_t(\mathbf{x}|c)q(c)\mathrm{d}c}{q_t(\mathbf{x})} \tag{S9}$$

$$= \frac{\int q_t(\mathbf{x}|c)q(c)\nabla_{\mathbf{x}} \log q_t(\mathbf{x}|c)\mathrm{d}c}{q_t(\mathbf{x})} \tag{S10}$$

$$= \int \frac{q_t(\mathbf{x}|c)q(c)}{q_t(\mathbf{x})}\nabla_{\mathbf{x}} \log q_t(\mathbf{x}|c)\mathrm{d}c \tag{S11}$$

$$= \mathbb{E}_{q_t(c|\mathbf{x})}[\nabla_{\mathbf{x}} \log q_t(\mathbf{x}|c)|\mathbf{x}]. \tag{S12}$$

Therefore, we have

$$\boldsymbol{\epsilon}_\theta(\mathbf{x}, t) = \mathbb{E}_{q_t(c|\mathbf{x})}[\boldsymbol{\epsilon}_\theta(\mathbf{x}, c, t)|\mathbf{x}] \tag{S13}$$

$$= \mathbb{E}_{q_t(c|\mathbf{x})}\left[\mathbb{E}_{q(\mathbf{x}_0|\mathbf{x},c)}\left[\frac{\mathbf{x} - \alpha_t\mathbf{x}_0}{\sigma_t}\right]|\mathbf{x}\right] \tag{S14}$$

$$= \mathbb{E}_{q_t(c,\mathbf{x}_0|\mathbf{x})}\left[\frac{\mathbf{x} - \alpha_t\mathbf{x}_0}{\sigma_t}|\mathbf{x}\right] \tag{S15}$$

$$= \frac{1}{\sigma_t}\mathbf{x} - \frac{\alpha_t}{\sigma_t}\mathbb{E}_{q_t(c,\mathbf{x}_0|\mathbf{x})}[\mathbf{x}_0|\mathbf{x}], \tag{S16}$$

and

$$\mathbb{E}_{\mathbf{x}}[\epsilon_\theta(\mathbf{x}, t)] = \frac{1}{\sigma_t}\mathbb{E}_{\mathbf{x}}[\mathbf{x}] - \frac{\alpha_t}{\sigma_t}\mathbb{E}_{\mathbf{x}}[\mathbb{E}_{q_t(c,\mathbf{x}_0|\mathbf{x})}[\mathbf{x}_0|\mathbf{x}]] \tag{S17}$$

$$= \frac{1}{\sigma_t}\mathbb{E}_{\mathbf{x}}[\mathbf{x}] - \frac{\alpha_t}{\sigma_t}\mathbb{E}_{c,\mathbf{x}_0,\mathbf{x}}[\mathbf{x}_0]. \tag{S18}$$

□

Then we start to prove Theorem 1.

*Proof of Theorem 1.* Similar to derivation in DDIM (Song et al., 2021), first rewrite $J_{\delta,\gamma}$ as below:

$$J_{\delta,\gamma} = \mathbb{E}\left[-\log p_{\theta,\gamma}(\mathbf{x}_0|\mathbf{x}_1, c) + \sum_{t=2}^{T} D_{KL}(q_\delta(\mathbf{x}_{t-1}|\mathbf{x}_t, \mathbf{x}_0, c)\|p_{\theta,\gamma}(\mathbf{x}_{t-1}|\mathbf{x}_t, c))\right] + C_1, \tag{S19}$$

in which $C_1$ is a constant not involving $\gamma$ and $\theta$.

Note that $\epsilon_\theta(\mathbf{x}_t, c, t) = \mathbb{E}_{q(\epsilon|\mathbf{x}_t, c)}[\epsilon|\mathbf{x}_t]$. Hence, for $t > 1$:

$$\mathbb{E}_{q(\mathbf{x}_t, \mathbf{x}_0|c)}[D_{KL}(q_\delta(\mathbf{x}_{t-1}|\mathbf{x}_t, \mathbf{x}_0, c)\|p_{\theta,\gamma}(\mathbf{x}_{t-1}|\mathbf{x}_t, c))] \tag{S20}$$

$$= \mathbb{E}_{q(\mathbf{x}_t, \mathbf{x}_0|c)}[D_{KL}(q_\delta(\mathbf{x}_{t-1}|\mathbf{x}_t, \mathbf{x}_0, c)\|q_\delta(\mathbf{x}_{t-1}|\mathbf{x}_t, \mathbf{f}_{\theta,\gamma}^t(\mathbf{x}_t, c), c))] \tag{S21}$$

$$\propto \mathbb{E}_{q(\mathbf{x}_t, \mathbf{x}_0|c)}[\|\mathbf{x}_0 - \mathbf{f}_{\theta,\gamma}^t(\mathbf{x}_t, c)\|_2^2] \tag{S22}$$

$$\propto \mathbb{E}_{\substack{\mathbf{x}_0 \sim q(\mathbf{x}_0|c) \\ \epsilon \sim \mathcal{N}(\mathbf{0},\mathbf{I}) \\ \mathbf{x}_t = \alpha_t\mathbf{x}_0 + \sigma_t\epsilon}}[\|\epsilon - (\gamma\epsilon_\theta(\mathbf{x}_t, c, t) + (1-\gamma)\epsilon_\theta(\mathbf{x}_t, t))\|_2^2] \tag{S23}$$

$$= \mathbb{E}_{\mathbf{x}_0,\epsilon}[\|\gamma(\epsilon - \epsilon_\theta(\mathbf{x}_t, c, t)) + (1-\gamma)(\epsilon - \epsilon_\theta(\mathbf{x}_t, t))\|_2^2] \tag{S24}$$

$$= \mathbb{E}_{\mathbf{x}_0,\epsilon}[\gamma^2\|\epsilon - \epsilon_\theta(\mathbf{x}_t, c, t)\|_2^2 + (1-\gamma)^2\|\epsilon - \epsilon_\theta(\mathbf{x}_t, t)\|_2^2$$
$$+ 2\gamma(1-\gamma)\mathbb{E}_{\mathbf{x}_0,\epsilon}[\langle\epsilon - \epsilon_\theta(\mathbf{x}_t, c, t), \epsilon - \epsilon_\theta(\mathbf{x}_t, t)\rangle] \tag{S25}$$

$$= \mathbb{E}_{\mathbf{x}_0,\epsilon}[\gamma^2\|\epsilon - \epsilon_\theta(\mathbf{x}_t, c, t)\|_2^2 + (1-\gamma)^2\|\epsilon - \epsilon_\theta(\mathbf{x}_t, t)\|_2^2$$
$$+ 2\gamma(1-\gamma)\mathbb{E}_{\mathbf{x}_0,\epsilon}[\langle\epsilon - \mathbb{E}_{q(\epsilon|\mathbf{x}_t, c)}[\epsilon|\mathbf{x}_t], \epsilon - \epsilon_\theta(\mathbf{x}_t, t)\rangle] \tag{S26}$$

$$= \mathbb{E}_{\mathbf{x}_0,\epsilon}[\gamma^2\|\epsilon - \epsilon_\theta(\mathbf{x}_t, c, t)\|_2^2 + (1-\gamma)^2\|\epsilon - \epsilon_\theta(\mathbf{x}_t, t)\|_2^2$$
$$+ 2\gamma(1-\gamma)\mathbb{E}_{\mathbf{x}_0,\epsilon}[\langle\epsilon - \epsilon, \epsilon - \epsilon_\theta(\mathbf{x}_t, t)\rangle] \tag{S27}$$

$$= \gamma^2\mathbb{E}_{\mathbf{x}_0,\epsilon}[\|\epsilon - \epsilon_\theta(\mathbf{x}_t, c, t)\|_2^2] + (1-\gamma)^2\mathbb{E}_{\mathbf{x}_0,\epsilon}[\|\epsilon - \epsilon_\theta(\mathbf{x}_t, t)\|_2^2], \tag{S28}$$

in which Eq. (S27) is from Lemma 1. As for $t = 1$ we have similar derivation:

$$\mathbb{E}_{q(\mathbf{x}_1, \mathbf{x}_0|c)}[-\log p_{\theta,\gamma}(\mathbf{x}_0|\mathbf{x}_1, c))] \tag{S29}$$

$$\propto \mathbb{E}_{q(\mathbf{x}_1, \mathbf{x}_0|c)}[\|\mathbf{x}_0 - \mathbf{f}_{\theta,\gamma}^t(\mathbf{x}_1, c)\|_2^2] + C_2 \tag{S30}$$

$$\propto \mathbb{E}_{\substack{\mathbf{x}_0 \sim q(\mathbf{x}_0|c) \\ \epsilon \sim \mathcal{N}(\mathbf{0},\mathbf{I}) \\ \mathbf{x}_1 = \alpha_1\mathbf{x}_0 + \sigma_1\epsilon}}[\|\epsilon - (\gamma\epsilon_\theta(\mathbf{x}_1, c, 1) + (1-\gamma)\epsilon_\theta(\mathbf{x}_1, 1))\|_2^2] + C_3 \tag{S31}$$

$$= \gamma^2\mathbb{E}_{\mathbf{x}_0,\epsilon}[\|\epsilon - \epsilon_\theta(\mathbf{x}_1, c, 1)\|_2^2] + (1-\gamma)^2\mathbb{E}_{\mathbf{x}_0,\epsilon}[\|\epsilon - \epsilon_\theta(\mathbf{x}_1, 1)\|_2^2] + C_3, \tag{S32}$$

in which $C_2$ and $C_3$ are constants not involving $\gamma$ and $\theta$. Given that CFG involves score matching using both conditional and unconditional distributions, and that $J_{\delta,\gamma}$ is proportional to the score matching objective up to a constant, we confirm the equivalence between $J_{\delta,\gamma}$ and objective of native DPM under CFG.

Note that in native PF-ODE, we have

$$\frac{d\mathbf{x}_t}{dt} = f(t)\mathbf{x}_t - \frac{1}{2}g^2(t)\nabla_{\mathbf{x}_t}\log q_t(\mathbf{x}_t|c), \tag{S33}$$

$$\mathbb{E}_{q_t(\mathbf{x}_t|c)}[\nabla_{\mathbf{x}_t}\log q_t(\mathbf{x}_t|c)] = \mathbb{E}_{q_t(\mathbf{x}_t|c)}[\mathbb{E}_{q_t(\mathbf{x}_0|\mathbf{x}_t, c)}[\nabla_{\mathbf{x}_t}\log q_t(\mathbf{x}_t|\mathbf{x}_0, c)]] \tag{S34}$$

$$= \mathbb{E}_{q_t(\mathbf{x}_0, \mathbf{x}_t|c)}[\nabla_{\mathbf{x}_t}\log q_t(\mathbf{x}_t|\mathbf{x}_0, c)] \tag{S35}$$

$$= 0, \tag{S36}$$

in which Eq. (S36) holds since forward diffusion process $q_t(\mathbf{x}_t|\mathbf{x}_0, c)$ is implemented by adding Gaussian noise. However, according to Eq. (10) and Lemma 2, we have

$$\mathbb{E}_{\mathbf{x}_t}[\nabla_{\mathbf{x}_t} \log q_{t,\gamma}(\mathbf{x}_t|c)] = \mathbb{E}_{\mathbf{x}_t}[\gamma \nabla_{\mathbf{x}_t} \log q_t(\mathbf{x}_t|c) + (1-\gamma)\nabla_{\mathbf{x}_t} \log q_t(\mathbf{x}_t)] \tag{S37}$$

$$= (1-\gamma)\mathbb{E}_{\mathbf{x}_t}[\nabla_{\mathbf{x}_t} \log q_t(\mathbf{x}_t)] \tag{S38}$$

$$= \frac{\gamma-1}{\sigma_t^2}(\mathbb{E}_{\mathbf{x}_t}[\mathbf{x}_t] - \alpha_t \mathbb{E}_{c,\mathbf{x}_0,\mathbf{x}_t}[\mathbf{x}_0]) \tag{S39}$$

$$= \frac{\gamma-1}{\sigma_t^2}(\mathbb{E}_{q_t(\mathbf{x}_t|c)}[\mathbf{x}_t] - \alpha_t \mathbb{E}_{q_0(\mathbf{x}_0,c)}[\mathbf{x}_0]). \tag{S40}$$

Note that $\mathbb{E}_{q_t(\mathbf{x}_t|c)}[\mathbf{x}_t] = \alpha_t \mathbb{E}_{q_0(\mathbf{x}_0|c)}[\mathbf{x}_0]$, and that $\mathbb{E}_{\mathbf{x}_0,c}[\mathbf{x}_0] = \int \mathbb{E}_{q_0(\mathbf{x}_0|c)}[\mathbf{x}_0]\mathrm{d}c$. Therefore when $\gamma \neq 1$, $\mathbb{E}_{\mathbf{x}_t}[\nabla_{\mathbf{x}_t} \log q_{t,\gamma}(\mathbf{x}_t|c)]$ is not guaranteed to be identical with 0. In other words, denoising with CFG cannot be expressed as a reciprocal of diffusion process with Gaussian noise. $\square$

### A.2 PROOF OF THEOREM 2

*Proof.* Given Eq. (9), for $\gamma > 1$, we have

$$\nabla_{\mathbf{x}_t} \log q_{t,\gamma}(\mathbf{x}_t|c) = \gamma \nabla_{\mathbf{x}_t} \log q_t(\mathbf{x}_t|c) + (1-\gamma)\nabla_{\mathbf{x}_t} \log q_t(\mathbf{x}_t) \tag{S41}$$

$$= -\gamma\frac{\mathbf{x}_t - c}{t+1} - (1-\gamma)\frac{\mathbf{x}_t}{t+2}, \tag{S42}$$

$$\frac{\mathrm{d}\mathbf{x}_t}{\mathrm{d}t} = -\frac{1}{2}\nabla_{\mathbf{x}_t} \log q_{t,\gamma}(\mathbf{x}_t|c) \tag{S43}$$

$$= \mathbf{x}_t\left(\frac{\gamma}{2(t+1)} + \frac{1-\gamma}{2(t+2)}\right) - c\frac{\gamma}{2(t+1)}. \tag{S44}$$

By variation of constants formula, we can analytically solve $q_{0,\gamma}^{\mathrm{deter}}(\mathbf{x}_0|c)$ in Eq. (S44).

$$\mathbf{x}_t = e^{\int_T^t \frac{\gamma}{2(s+1)} + \frac{1-\gamma}{2(s+2)}\mathrm{d}s}\left(C - \int_T^t c\frac{\gamma}{2(s+1)}e^{-\int_s^t \frac{\gamma}{2(r+1)} + \frac{1-\gamma}{2(r+2)}\mathrm{d}r}\mathrm{d}s\right) \tag{S45}$$

$$= (t+1)^{\frac{\gamma}{2}}(t+2)^{\frac{1-\gamma}{2}}\left(C - c\frac{\gamma}{2}\int_T^t (s+1)^{-\frac{\gamma+2}{2}}(s+2)^{-\frac{1-\gamma}{2}}\mathrm{d}s\right), \tag{S46}$$

in which $C$ is a constant to determine. Let $t = T$, we can see that

$$C = \frac{\mathbf{x}_T}{(T+1)^{\frac{\gamma}{2}}(T+2)^{\frac{1-\gamma}{2}}}. \tag{S47}$$

Therefore, we achieve the closed-form formula for $q_{0,\gamma}^{\mathrm{deter}}(\mathbf{x}_0|c)$ as below:

$$\mathbf{x}_0 = 2^{\frac{1-\gamma}{2}}\left(\frac{\mathbf{x}_T}{(T+1)^{\frac{\gamma}{2}}(T+2)^{\frac{1-\gamma}{2}}} + c\frac{\gamma}{2}\int_0^T (s+1)^{-\frac{\gamma+2}{2}}(s+2)^{-\frac{1-\gamma}{2}}\mathrm{d}s\right). \tag{S48}$$

Since $q_T(\mathbf{x}_T|c) \sim \mathcal{N}(c, T+1)$, we can deduce that

$$q_{0,\gamma}^{\mathrm{deter}}(\mathbf{x}_0|c) \sim \mathcal{N}\left(c\phi(\gamma, T), 2^{1-\gamma}\frac{T+1}{(T+1)^{\gamma}(T+2)^{1-\gamma}}\right), \tag{S49}$$

in which

$$\phi(\gamma, T) = 2^{\frac{1-\gamma}{2}}\left(\frac{1}{(T+1)^{\frac{\gamma}{2}}(T+2)^{\frac{1-\gamma}{2}}} + \frac{\gamma}{2}\int_0^T (s+1)^{-\frac{\gamma+2}{2}}(s+2)^{-\frac{1-\gamma}{2}}\mathrm{d}s\right). \tag{S50}$$

It is obvious that

$$\phi(\gamma) = 2^{\frac{1-\gamma}{2}}\frac{\gamma}{2}\int_0^{+\infty} (s+1)^{-\frac{\gamma+2}{2}}(s+2)^{-\frac{1-\gamma}{2}}\mathrm{d}s, \tag{S51}$$

$$\lim_{T\to+\infty} \frac{T+1}{(T+1)^{\gamma}(T+2)^{1-\gamma}} = 1. \tag{S52}$$

Then it is suffices to calculate $\phi(\gamma)$ for all $\gamma > 1$. First note that

$$\phi(1) = \frac{1}{2} \int_0^{+\infty} (s+1)^{-\frac{3}{2}} \mathrm{d}s = 1, \tag{S53}$$

$$\phi(3) = 2^{-1} \frac{3}{2} \int_0^{+\infty} (s+1)^{-\frac{5}{2}} (s+2) \mathrm{d}s = 2, \tag{S54}$$

$$\phi(5) = 2^{-2} \frac{5}{2} \int_0^{+\infty} (s+1)^{-\frac{7}{2}} (s+2)^2 \mathrm{d}s = \frac{7}{3}. \tag{S55}$$

For $\gamma > 1$, denote by $I(\gamma)$ with

$$I(\gamma) = \int_0^{+\infty} (s+1)^{-\frac{\gamma+2}{2}} (s+2)^{-\frac{1-\gamma}{2}} \mathrm{d}s. \tag{S56}$$

Note that for $\gamma > 1$ we have

$$I(\gamma) = \int_0^{+\infty} (s+1)^{-\frac{\gamma+2}{2}} (s+2)^{-\frac{1-\gamma}{2}} \mathrm{d}s \tag{S57}$$

$$= \frac{2}{\gamma+1} \int_0^{+\infty} (s+1)^{-\frac{\gamma+2}{2}} \mathrm{d}(s+2)^{\frac{\gamma+1}{2}} \tag{S58}$$

$$= \frac{2}{\gamma+1} \left( (s+1)^{-\frac{\gamma+1}{2}} (s+2)^{\frac{\gamma+1}{2}} \Big|_0^{+\infty} + \frac{\gamma+2}{2} \int_0^{+\infty} (s+1)^{-\frac{\gamma+4}{2}} (s+2)^{\frac{\gamma+1}{2}} \mathrm{d}s \right) \tag{S59}$$

$$= \frac{2}{\gamma+1} \left( \frac{\gamma+2}{2} I(\gamma+2) - 2^{\frac{\gamma+1}{2}} \right). \tag{S60}$$

Therefore, for $\gamma > 1$ we have

$$\phi(\gamma) = \frac{2\gamma}{\gamma+1} (\phi(\gamma+2) - 1), \tag{S61}$$

$$\phi(\gamma+2) = 1 + \frac{\gamma+1}{2\gamma} \phi(\gamma). \tag{S62}$$

From Eqs. (S53) to (S55) we have

$$I(1) = 2, \quad I(3) = \frac{8}{3}, \quad I(5) = \frac{56}{15}. \tag{S63}$$

For $\gamma \in [1,3]$, by Cauchy-Schwarz inequality with $p \in [0,1]$, we have

$$\left( I(\gamma) \right)^p \left( I(5) \right)^{1-p} \tag{S64}$$

$$= \left( \int_0^{+\infty} (s+1)^{-\frac{\gamma+2}{2}} (s+2)^{-\frac{1-\gamma}{2}} \mathrm{d}s \right)^p \left( \int_0^{+\infty} (s+1)^{-\frac{7}{2}} (s+2)^2 \mathrm{d}s \right)^{1-p} \tag{S65}$$

$$\geqslant \int_0^{+\infty} \left( (s+1)^{-\frac{\gamma+2}{2}} (s+2)^{-\frac{1-\gamma}{2}} \right)^p \left( (s+1)^{-\frac{7}{2}} (s+2)^2 \right)^{1-p} \mathrm{d}s \tag{S66}$$

$$= \int_0^{+\infty} (s+1)^{-\frac{\gamma p - 5p + 7}{2}} (s+2)^{-\frac{5p - \gamma p - 4}{2}} \mathrm{d}s. \tag{S67}$$

Let $p = \frac{2}{5-\gamma} \in [0,1]$ for $\gamma \in [1,3]$, from Eq. (S67) we have

$$I(\gamma) \geqslant \left( I(3) \right)^{\frac{5-\gamma}{2}} \left( I(5) \right)^{\frac{\gamma-3}{2}} = \left( \frac{8}{3} \right)^{\frac{5-\gamma}{2}} \left( \frac{56}{15} \right)^{\frac{\gamma-3}{2}}. \tag{S68}$$

Therefore for $\gamma \in [1,3]$, we have

$$\phi(\gamma) \geqslant 2^{\frac{1-\gamma}{2}} \frac{\gamma}{2} \left( \frac{8}{3} \right)^{\frac{5-\gamma}{2}} \left( \frac{56}{15} \right)^{\frac{\gamma-3}{2}} = \gamma \frac{7}{15} \left( \frac{10}{7} \right)^{\frac{5-\gamma}{2}} =: h_1(\gamma) \tag{S69}$$

Since $\frac{1}{\gamma} - \frac{1}{2}\log\frac{10}{7} > 0$ for $\gamma \in [1,3]$, $h_1(\gamma)$ increases monotonically on $\in [1,3]$ and $h_1(1) = \frac{20}{21}$, $h_1(3) = 2$. Similarly, for $\gamma \in [3,5]$, by Cauchy-Schwarz inequality with $p \in [0,1]$, we have

$$\Big(I(1)\Big)^{1-p}\Big(I(\gamma)\Big)^p \tag{S70}$$

$$= \left(\int_0^{+\infty} (s+1)^{-\frac{3}{2}}\,\mathrm{d}s\right)^{1-p} \left(\int_0^{+\infty} (s+1)^{-\frac{\gamma+2}{2}}(s+2)^{-\frac{1-\gamma}{2}}\,\mathrm{d}s\right)^p \tag{S71}$$

$$\geqslant \int_0^{+\infty} \Big((s+1)^{-\frac{3}{2}}\Big)^{1-p}\Big((s+1)^{-\frac{\gamma+2}{2}}(s+2)^{-\frac{1-\gamma}{2}}\Big)^p\,\mathrm{d}s \tag{S72}$$

$$= \int_0^{+\infty} (s+1)^{-\frac{3-p+\gamma p}{2}}(s+2)^{-\frac{p-\gamma p}{2}}\,\mathrm{d}s. \tag{S73}$$

Let $p = \frac{2}{\gamma-1} \in [0,1]$ for $\gamma \in [3,5]$, from Eq. (S73) we have

$$I(\gamma) \geqslant \Big(I(1)\Big)^{\frac{3-\gamma}{2}}\Big(I(3)\Big)^{\frac{\gamma-1}{2}} = 2^{\frac{3-\gamma}{2}}\left(\frac{8}{3}\right)^{\frac{\gamma-1}{2}}. \tag{S74}$$

Therefore for $\gamma \in [3,5]$

$$\phi(\gamma) \geqslant 2^{\frac{1-\gamma}{2}}\frac{\gamma}{2}2^{\frac{3-\gamma}{2}}\left(\frac{8}{3}\right)^{\frac{\gamma-1}{2}} = \gamma\left(\frac{2}{3}\right)^{\frac{\gamma-1}{2}} =: h_2(\gamma) \tag{S75}$$

It is easy to see that $h_2(\gamma) \geqslant 2$ for $\gamma \in [3,5]$. Then by mathematical induction and Eq. (S62), we have $\phi(\gamma) \geqslant 2$ for all $\gamma \geqslant 3$. Specially, we have

$$\lim_{\gamma \to +\infty} \phi(\gamma) = 2. \tag{S76}$$

And specifically, for $\gamma \in \mathbb{N}$, $\gamma > 1$, we analytically calculate $\phi(\gamma, T)$ for $\gamma = 2n+1$ and $\gamma = 2n$, respectively. First let $\gamma = 2n+1$, $n \in \mathbb{N}$. We can see that

$$(s+2)^{-\frac{1-\gamma}{2}} = (s+2)^n = \sum_{k=0}^n C_n^k(s+1)^k. \tag{S77}$$

$$\int_0^T (s+1)^{-\frac{\gamma+2}{2}}(s+2)^{-\frac{1-\gamma}{2}}\,\mathrm{d}s = \int_0^T (s+1)^{-\frac{\gamma+2}{2}}\left(\sum_{k=0}^n C_n^k(s+1)^k\right)\mathrm{d}s \tag{S78}$$

$$= \sum_{k=0}^n \left(C_n^k \int_0^T (s+1)^{\frac{2k-\gamma-2}{2}}\,\mathrm{d}s\right) \tag{S79}$$

$$= \sum_{k=0}^n \left(C_n^k \frac{2}{2k-\gamma}\Big((T+1)^{\frac{2k-\gamma}{2}} - 1\Big)\right). \tag{S80}$$

Since $2k - \gamma < 0$ for $k = 0, 1, \cdots, n$, we have $(T+1)^{\frac{2k-\gamma}{2}} \to 0$ as $T$ goes to infinity and hence

$$\phi(\gamma, T) \tag{S81}$$

$$= 2^{\frac{1-\gamma}{2}}\left(\frac{1}{(T+1)^{\frac{\gamma}{2}}(T+2)^{\frac{1-\gamma}{2}}} + \frac{\gamma}{2}\int_0^T (s+1)^{-\frac{\gamma+2}{2}}(s+2)^{-\frac{1-\gamma}{2}}\,\mathrm{d}s\right) \tag{S82}$$

$$= 2^{\frac{1-\gamma}{2}}\left(\frac{1}{(T+1)^{\frac{\gamma}{2}}(T+2)^{\frac{1-\gamma}{2}}} + \frac{\gamma}{2}\left(\sum_{k=0}^n C_n^k\frac{2}{2k-\gamma}\Big((T+1)^{\frac{2k-\gamma}{2}} - 1\Big)\right)\right) \tag{S83}$$

$$= 2^{\frac{1-\gamma}{2}}\left(\frac{1}{(T+1)^{\frac{\gamma}{2}}(T+2)^{\frac{1-\gamma}{2}}} + \left(\sum_{k=0}^n C_n^k\frac{2n+1}{2n-2k+1}\Big(1 - (T+1)^{\frac{2k-\gamma}{2}}\Big)\right)\right). \tag{S84}$$

When $T \to +\infty$, we have

$$\phi(2n+1) = 2^{-n}\left(\sum_{k=0}^n C_n^k\frac{2n+1}{2n-2k+1}\right). \tag{S85}$$

Then let $\gamma = 2n$, $n \in \mathbb{N}$, and $n \geqslant 1$. We have

$$\int (s+1)^{-\frac{\gamma+2}{2}}(s+2)^{-\frac{1-\gamma}{2}}\mathrm{d}s \tag{S86}$$

$$= \int 2(s+1)^{-n-1}(\sqrt{s+2})^{2n}\mathrm{d}\sqrt{s+2} \tag{S87}$$

$$= \int 2(u^2-1)^{-n-1}u^{2n}\mathrm{d}u \tag{S88}$$

$$= -\frac{1}{n}u^{2n-1}(u^2-1)^{-n} + \frac{2n-1}{n}\int (u^2-1)^{-n}u^{2n-2}\mathrm{d}u, \tag{S89}$$

in which Eq. (S88) is due to integration by substitution with $u = \sqrt{s+2} > 1$, and Eq. (S89) is due to integration by parts. Denote by $I_n$ with

$$I_n = \int 2(u^2-1)^{-n-1}u^{2n}\mathrm{d}u, \quad n \geqslant 0, \tag{S90}$$

then we have

$$I_n = -\frac{1}{n}u^{2n-1}(u^2-1)^{-n} + \frac{2n-1}{2n}I_{n-1}, \quad n \geqslant 1. \tag{S91}$$

For $n \geqslant 1$, let $I_n = \frac{(2n-1)!!}{(2n)!!}A_n$, then we have

$$\frac{(2n-1)!!}{(2n)!!}A_n = -\frac{1}{n}u^{2n-1}(u^2-1)^{-n} + \frac{2n-1}{2n}\frac{(2n-3)!!}{(2n-2)!!}A_{n-1}, \quad n \geqslant 2, \tag{S92}$$

$$A_n = A_{n-1} - \frac{1}{n}\frac{(2n)!!}{(2n-1)!!}u^{2n-1}(u^2-1)^{-n}, \quad n \geqslant 2. \tag{S93}$$

Therefore for $n \geqslant 2$, we have

$$A_n = A_1 - \sum_{k=2}^{n}\frac{1}{k}\frac{(2k)!!}{(2k-1)!!}u^{2k-1}(u^2-1)^{-k}, \tag{S94}$$

and

$$I_n = \begin{cases} -\dfrac{u}{u^2-1} + \dfrac{1}{2}\log\dfrac{u-1}{u+1}, & n = 1, \\[3mm] \dfrac{(2n-1)!!}{(2n)!!}\left(-\dfrac{2u}{u^2-1} + \log\dfrac{u-1}{u+1} - \sum_{k=2}^{n}\dfrac{1}{k}\dfrac{(2k)!!}{(2k-1)!!}\dfrac{u^{2k-1}}{(u^2-1)^k}\right), & n \geqslant 2. \end{cases} \tag{S95}$$

Therefore, for $\gamma = 2$ we have

$$\phi(\gamma, T) = 2^{\frac{1-\gamma}{2}}\left(\frac{1}{(T+1)^{\frac{\gamma}{2}}(T+2)^{\frac{1-\gamma}{2}}} + \frac{\gamma}{2}\int_0^T (s+1)^{-\frac{\gamma+2}{2}}(s+2)^{-\frac{1-\gamma}{2}}\mathrm{d}s\right) \tag{S96}$$

$$= 2^{\frac{1-\gamma}{2}}\frac{1}{(T+1)^{\frac{\gamma}{2}}(T+2)^{\frac{1-\gamma}{2}}}$$

$$\quad - 2^{\frac{1-\gamma}{2}}\frac{\gamma}{2}\left(\frac{\sqrt{T+2}}{T+1} - \sqrt{2}\right)$$

$$\quad + 2^{\frac{1-\gamma}{2}}\frac{\gamma}{2}\frac{1}{2}\left(\log\frac{\sqrt{T+2}-1}{\sqrt{T+2}+1} - \log\frac{\sqrt{2}-1}{\sqrt{2}+1}\right), \tag{S97}$$

and for $\gamma \geqslant 4$ we have

$$\phi(\gamma, T) \tag{S98}$$

$$= 2^{\frac{1-\gamma}{2}} \left( \frac{1}{(T+1)^{\frac{\gamma}{2}}(T+2)^{\frac{1-\gamma}{2}}} + \frac{\gamma}{2} \int_0^T (s+1)^{-\frac{\gamma+2}{2}}(s+2)^{-\frac{1-\gamma}{2}} \mathrm{d}s \right) \tag{S99}$$

$$= 2^{\frac{1-\gamma}{2}} \frac{1}{(T+1)^{\frac{\gamma}{2}}(T+2)^{\frac{1-\gamma}{2}}}$$

$$\qquad - 2^{\frac{1-\gamma}{2}} \frac{\gamma}{2} \frac{(2n-1)!!}{(2n)!!} \left( \frac{2\sqrt{T+2}}{T+1} - 2\sqrt{2} \right)$$

$$\qquad + 2^{\frac{1-\gamma}{2}} \frac{\gamma}{2} \frac{(2n-1)!!}{(2n)!!} \left( \log \frac{\sqrt{T+2}-1}{\sqrt{T+2}+1} - \log \frac{\sqrt{2}-1}{\sqrt{2}+1} \right)$$

$$\qquad - 2^{\frac{1-\gamma}{2}} \frac{\gamma}{2} \frac{(2n-1)!!}{(2n)!!} \left( \sum_{k=2}^n \frac{1}{k} \frac{(2k)!!}{(2k-1)!!} \left( \frac{(T+2)^{\frac{2k-1}{2}}}{(T+1)^k} - 2^{\frac{2k-1}{2}} \right) \right). \tag{S100}$$

When $T \to +\infty$, we have

$$\phi(2n) = \begin{cases} 2^{\frac{1}{2}-n} n \left( \sqrt{2} - \frac{1}{2} \log \frac{\sqrt{2}-1}{\sqrt{2}+1} \right), & n = 1, \\[2ex] \frac{(2n-1)!!\sqrt{2}n}{(2n)!!2^n} \left( \left( \sum_{k=2}^n \frac{1}{k} \frac{(2k)!!}{(2k-1)!!} 2^{k-\frac{1}{2}} \right) + 2\sqrt{2} - \log \frac{\sqrt{2}-1}{\sqrt{2}+1} \right), & n \geqslant 2. \end{cases} \tag{S101}$$

$\square$

## A.3 Proof of Theorem 3

*Proof.* We first write the closed-form expressions of DDIM sampler as below:

$$\mathbf{x}_{t-1} = \frac{\alpha_{t-1}}{\alpha_t} \mathbf{x}_t + (\sigma_{t-1} - \frac{\alpha_{t-1}}{\alpha_t} \sigma_t) \boldsymbol{\epsilon}_\theta(\mathbf{x}_t, c, t), \tag{S102}$$

$$\tilde{\mathbf{x}}_{t-1} = \frac{\alpha_{t-1}}{\alpha_t} \tilde{\mathbf{x}}_t + (\sigma_{t-1} - \frac{\alpha_{t-1}}{\alpha_t} \sigma_t)(\gamma_1 \boldsymbol{\epsilon}_\theta(\tilde{\mathbf{x}}_t, c, t) + \gamma_0 \boldsymbol{\epsilon}_\theta(\tilde{\mathbf{x}}_t, t)). \tag{S103}$$

Then we have

$$\Delta_{t-1} = \mathbb{E}_{\mathbf{x}_t}[\mathbf{x}_{t-1}] - \mathbb{E}_{\tilde{\mathbf{x}}_t}[\tilde{\mathbf{x}}_{t-1}] \tag{S104}$$

$$= \frac{\alpha_{t-1}}{\alpha_t} (\mathbb{E}_{\mathbf{x}_t}[\mathbf{x}_t] - \mathbb{E}_{\tilde{\mathbf{x}}_t}[\tilde{\mathbf{x}}_t])$$

$$\qquad + (\sigma_{t-1} - \frac{\alpha_{t-1}}{\alpha_t} \sigma_t)(\mathbb{E}_{\mathbf{x}_t}[\boldsymbol{\epsilon}_\theta(\mathbf{x}_t, c)] - \mathbb{E}_{\tilde{\mathbf{x}}_t}[\gamma_1 \boldsymbol{\epsilon}_\theta(\tilde{\mathbf{x}}_t, c, t) + \gamma_0 \boldsymbol{\epsilon}_\theta(\tilde{\mathbf{x}}_t, t)]). \tag{S105}$$

Note that

$$\boldsymbol{\epsilon}_\theta(\mathbf{x}_t, c, t) = \mathbb{E}_{q(\mathbf{x}_0|\mathbf{x}_t,c)} \left[ \frac{\mathbf{x}_t - \alpha_t \mathbf{x}_0}{\sigma_t} | \mathbf{x}_t \right], \quad \boldsymbol{\epsilon}_\theta(\tilde{\mathbf{x}}_t, c, t) = \mathbb{E}_{q(\mathbf{x}_0|\tilde{\mathbf{x}}_t,c)} \left[ \frac{\tilde{\mathbf{x}}_t - \alpha_t \mathbf{x}_0}{\sigma_t} | \tilde{\mathbf{x}}_t \right]. \tag{S106}$$

Therefore, by $q_t(\mathbf{x}_t|c) = \int q_0(\mathbf{x}_0|c) q_{0t}(\mathbf{x}_t|\mathbf{x}_0, c) \mathrm{d}\mathbf{x}_0$ and Lemma 1 we have

$$\mathbb{E}_{\mathbf{x}_t}[\boldsymbol{\epsilon}_\theta(\mathbf{x}_t, c, t)] = \mathbb{E}_{\mathbf{x}_0, \mathbf{x}_t} \left[ \frac{\mathbf{x}_t - \alpha_t \mathbf{x}_0}{\sigma_t} \right] = \frac{1}{\sigma_t} \mathbb{E}_{\mathbf{x}_t}[\mathbf{x}_t] - \frac{\alpha_t}{\sigma_t} \mathbb{E}_{\mathbf{x}_0}[\mathbf{x}_0]. \tag{S107}$$

Similarly, by $p_{\theta, \gamma_1, \gamma_0}(\tilde{\mathbf{x}}_t|c) = \int q_0(\mathbf{x}_0|c) q_{0T}(\mathbf{x}_T|\mathbf{x}_0, c) p_{\theta, \gamma_1, \gamma_0}(\tilde{\mathbf{x}}_t|\mathbf{x}_T, c) \mathrm{d}\mathbf{x}_0 \mathrm{d}\mathbf{x}_T$ we have

$$\mathbb{E}_{\tilde{\mathbf{x}}_t}[\boldsymbol{\epsilon}_\theta(\tilde{\mathbf{x}}_t, c, t)] = \mathbb{E}_{\mathbf{x}_0, \tilde{\mathbf{x}}_t} \left[ \frac{\tilde{\mathbf{x}}_t - \alpha_t \mathbf{x}_0}{\sigma_t} \right] = \frac{1}{\sigma_t} \mathbb{E}_{\tilde{\mathbf{x}}_t}[\tilde{\mathbf{x}}_t] - \frac{\alpha_t}{\sigma_t} \mathbb{E}_{\mathbf{x}_0}[\mathbf{x}_0]. \tag{S108}$$

Then we can simplify $\Delta_t$ as below:

$$\Delta_{t-1} = \frac{\alpha_{t-1}}{\alpha_t}\Delta_t + (\sigma_{t-1} - \frac{\alpha_{t-1}}{\alpha_t}\sigma_t)(\frac{1}{\sigma_t}\Delta_t - \mathbb{E}_{\tilde{\mathbf{x}}_t}[(\gamma_1 - 1)\boldsymbol{\epsilon}_\theta(\tilde{\mathbf{x}}_t, c, t) + \gamma_0\boldsymbol{\epsilon}_\theta(\tilde{\mathbf{x}}_t, t)]) \quad \text{(S109)}$$

$$= \frac{\sigma_{t-1}}{\sigma_t}\Delta_t - (\sigma_{t-1} - \frac{\alpha_{t-1}}{\alpha_t}\sigma_t)\mathbb{E}_{\tilde{\mathbf{x}}_t}[(\gamma_1 - 1)\boldsymbol{\epsilon}_\theta(\tilde{\mathbf{x}}_t, c, t) + \gamma_0\boldsymbol{\epsilon}_\theta(\tilde{\mathbf{x}}_t, t)]). \quad \text{(S110)}$$

$\Delta_t = 0$ implies that $\mathbb{E}_{q_t(\mathbf{x}_t|c)}[\mathbf{x}_t] = \mathbb{E}_{p_{\theta,\gamma_1,\gamma_0}(\tilde{\mathbf{x}}_t|c)}[\tilde{\mathbf{x}}_t]$. Therefore, by Eqs. (S107) and (S108) we have $\mathbb{E}_{\mathbf{x}_t}[\boldsymbol{\epsilon}_\theta(\mathbf{x}_t, c, t)] = \mathbb{E}_{\tilde{\mathbf{x}}_t}[\boldsymbol{\epsilon}_\theta(\tilde{\mathbf{x}}_t, c, t)]$. According to Lemma 2 and by calculating the expectation over $\mathbf{x}_t$ and $\tilde{\mathbf{x}}_t$ respectively, we have

$$\mathbb{E}_{\tilde{\mathbf{x}}_t}[\boldsymbol{\epsilon}_\theta(\tilde{\mathbf{x}}_t, t)] = \frac{1}{\sigma_t}\mathbb{E}_{\tilde{\mathbf{x}}_t}[\tilde{\mathbf{x}}_t] - \frac{\alpha_t}{\sigma_t}\mathbb{E}_{c,\mathbf{x}_0,\tilde{\mathbf{x}}_t}[\mathbf{x}_0] = \frac{1}{\sigma_t}\mathbb{E}_{\tilde{\mathbf{x}}_t}[\tilde{\mathbf{x}}_t] - \frac{\alpha_t}{\sigma_t}\mathbb{E}_{c,\mathbf{x}_0}[\mathbf{x}_0], \quad \text{(S111)}$$

$$\mathbb{E}_{\mathbf{x}_t}[\boldsymbol{\epsilon}_\theta(\mathbf{x}_t, t)] = \frac{1}{\sigma_t}\mathbb{E}_{\tilde{\mathbf{x}}_t}[\mathbf{x}_t] - \frac{\alpha_t}{\sigma_t}\mathbb{E}_{c,\mathbf{x}_0,\mathbf{x}_t}[\mathbf{x}_0] = \frac{1}{\sigma_t}\mathbb{E}_{\mathbf{x}_t}[\mathbf{x}_t] - \frac{\alpha_t}{\sigma_t}\mathbb{E}_{c,\mathbf{x}_0}[\mathbf{x}_0]. \quad \text{(S112)}$$

Since $\Delta_t = 0$, we have $\mathbb{E}_{\mathbf{x}_t}[\boldsymbol{\epsilon}_\theta(\mathbf{x}_t, t)] = \mathbb{E}_{\tilde{\mathbf{x}}_t}[\boldsymbol{\epsilon}_\theta(\tilde{\mathbf{x}}_t, t)]$, and thus

$$\Delta_{t-1} = -(\sigma_{t-1} - \frac{\alpha_{t-1}}{\alpha_t}\sigma_t)\mathbb{E}_{\mathbf{x}_t}[(\gamma_1 - 1)\boldsymbol{\epsilon}_\theta(\mathbf{x}_t, c, t) + \gamma_0\boldsymbol{\epsilon}_\theta(\mathbf{x}_t, t)]). \quad \text{(S113)}$$

$\square$

### A.4 PROOF OF THEOREM 4

*Proof.* Given Eq. (27), for any $\gamma_1$ and $\gamma_0$, we have

$$\nabla_{\mathbf{x}_t}\log q_{t,\gamma_1,\gamma_0}(\mathbf{x}_t|c) = \gamma_1\nabla_{\mathbf{x}_t}\log q_t(\mathbf{x}_t|c) + \gamma_0\nabla_{\mathbf{x}_t}\log q_t(\mathbf{x}_t) \quad \text{(S114)}$$

$$= -\gamma_1\frac{\mathbf{x}_t - c}{t+1} - \gamma_0\frac{\mathbf{x}_t}{t+2}, \quad \text{(S115)}$$

$$\frac{\mathrm{d}\mathbf{x}_t}{\mathrm{d}t} = -\frac{1}{2}\nabla_{\mathbf{x}_t}\log q_{t,\gamma_1,\gamma_0}(\mathbf{x}_t|c) \quad \text{(S116)}$$

$$= \mathbf{x}_t\left(\frac{\gamma_1}{2(t+1)} + \frac{\gamma_0}{2(t+2)}\right) - c\frac{\gamma_1}{2(t+1)}. \quad \text{(S117)}$$

By variation of constants formula, we can analytically solve $q_{0,\gamma_1,\gamma_0}^{\text{deter}}(\mathbf{x}_0|c)$ in Eq. (S117).

$$\mathbf{x}_t = e^{\int_T^t \frac{\gamma_1}{2(s+1)} + \frac{\gamma_0}{2(s+2)}\mathrm{d}s}\left(C - \int_T^t c\frac{\gamma_1}{2(s+1)}e^{-\int_s^t \frac{\gamma_1}{2(r+1)} + \frac{\gamma_0}{2(r+2)}\mathrm{d}r}\mathrm{d}s\right) \quad \text{(S118)}$$

$$= (t+1)^{\frac{\gamma_1}{2}}(t+2)^{\frac{\gamma_0}{2}}\left(C - c\frac{\gamma_1}{2}\int_T^t (s+1)^{-\frac{\gamma_1+2}{2}}(s+2)^{-\frac{\gamma_0}{2}}\mathrm{d}s\right), \quad \text{(S119)}$$

in which $C$ is a constant to determine. Let $t = T$, we can see that

$$C = \frac{\mathbf{x}_T}{(T+1)^{\frac{\gamma_1}{2}}(T+2)^{\frac{\gamma_0}{2}}}. \quad \text{(S120)}$$

Therefore, we achieve the closed-form formula for $q_{0,\gamma}^{\text{deter}}(\mathbf{x}_0|c)$ as below:

$$\mathbf{x}_0 = 2^{\frac{\gamma_0}{2}}\left(\frac{\mathbf{x}_T}{(T+1)^{\frac{\gamma_1}{2}}(T+2)^{\frac{\gamma_0}{2}}} + c\frac{\gamma_1}{2}\int_0^T (s+1)^{-\frac{\gamma_1+2}{2}}(s+2)^{-\frac{\gamma_0}{2}}\mathrm{d}s\right). \quad \text{(S121)}$$

Since $q_T(\mathbf{x}_T|c) \sim \mathcal{N}(c, T+1)$, we can deduce that

$$\text{var}_{q_{0,\gamma_1,\gamma_0}^{\text{deter}}(\mathbf{x}_0|c)}[\mathbf{x}_0] = 2^{\gamma_0}(T+1)^{1-\gamma_1}(T+2)^{-\gamma_0}. \quad \text{(S122)}$$

$\square$

## A.5 Proof of Theorem 5

*Proof.* According to Eqs. (29) and (30), we can write the variational lower bound of $p_{\theta,\gamma_1,\gamma_0}(\mathbf{x}_{0:T}|c)$ as below:

$$J_{\delta,\gamma_1,\gamma_0} = \mathbb{E}_{q_\delta(\mathbf{x}_{0:T}|c)}[\log q_\delta(\mathbf{x}_{1:T}|\mathbf{x}_0,c) - \log p_{\theta,\gamma_1,\gamma_0}(\mathbf{x}_{0:T}|c)] \tag{S123}$$

$$= \mathbb{E}\left[-\log p_{\theta,\gamma_1,\gamma_0}(\mathbf{x}_0|\mathbf{x}_1,c)\right]$$

$$+ \mathbb{E}\left[\sum_{t=2}^T D_{KL}(q_\delta(\mathbf{x}_{t-1}|\mathbf{x}_t,\mathbf{x}_0,c)\|p_{\theta,\gamma_1,\gamma_0}(\mathbf{x}_{t-1}|\mathbf{x}_t,c))\right]$$

$$+ C_1, \tag{S124}$$

in which $C_1$ is a constant not involving $\gamma_1$, $\gamma_0$, and $\theta$.

Note that $\boldsymbol{\epsilon}_\theta(\mathbf{x}_t,c,t) = \mathbb{E}_{q(\boldsymbol{\epsilon}|\mathbf{x}_t,c)}[\boldsymbol{\epsilon}|\mathbf{x}_t]$. Hence, for $t > 1$:

$$\mathbb{E}_{q(\mathbf{x}_t,\mathbf{x}_0|c)}[D_{KL}(q_\delta(\mathbf{x}_{t-1}|\mathbf{x}_t,\mathbf{x}_0,c)\|p_{\theta,\gamma_1,\gamma_0}(\mathbf{x}_{t-1}|\mathbf{x}_t,c))] \tag{S125}$$

$$= \mathbb{E}_{q(\mathbf{x}_t,\mathbf{x}_0|c)}[D_{KL}(q_\delta(\mathbf{x}_{t-1}|\mathbf{x}_t,\mathbf{x}_0,c)\|q_\delta(\mathbf{x}_{t-1}|\mathbf{x}_t,\mathbf{f}_{\theta,\gamma_1,\gamma_0}^t(\mathbf{x}_t,c),c))] \tag{S126}$$

$$\propto \mathbb{E}_{q(\mathbf{x}_t,\mathbf{x}_0|c)}[\|\mathbf{x}_0 - \mathbf{f}_{\theta,\gamma_1,\gamma_0}^t(\mathbf{x}_t,c)\|_2^2] \tag{S127}$$

$$\propto \mathbb{E}_{\substack{\mathbf{x}_0\sim q(\mathbf{x}_0|c)\\\boldsymbol{\epsilon}\sim\mathcal{N}(\mathbf{0},\mathbf{I})\\\mathbf{x}_t=\alpha_t\mathbf{x}_0+\sigma_t\boldsymbol{\epsilon}}}[\|\boldsymbol{\epsilon} - (\gamma_1\boldsymbol{\epsilon}_\theta(\mathbf{x}_t,c,t) + \gamma_0\boldsymbol{\epsilon}_\theta(\mathbf{x}_t,t))\|_2^2] \tag{S128}$$

$$= \mathbb{E}_{\mathbf{x}_0,\boldsymbol{\epsilon}}[\|\boldsymbol{\epsilon}\|_2^2 + \|\gamma_1\boldsymbol{\epsilon}_\theta(\mathbf{x}_t,c,t) + \gamma_0\boldsymbol{\epsilon}_\theta(\mathbf{x}_t,t)\|_2^2$$

$$- 2\mathbb{E}_{\mathbf{x}_0,\boldsymbol{\epsilon}}[\langle\boldsymbol{\epsilon},\gamma_1\boldsymbol{\epsilon}_\theta(\mathbf{x}_t,c,t) + \gamma_0\boldsymbol{\epsilon}_\theta(\mathbf{x}_t,t)\rangle] \tag{S129}$$

$$= \mathbb{E}_{\mathbf{x}_0,\boldsymbol{\epsilon}}[\|\boldsymbol{\epsilon}\|_2^2 + \|\gamma_1\boldsymbol{\epsilon}_\theta(\mathbf{x}_t,c,t) + \gamma_0\boldsymbol{\epsilon}_\theta(\mathbf{x}_t,t)\|_2^2$$

$$- 2\mathbb{E}_{\mathbf{x}_0,\boldsymbol{\epsilon}}[\langle\mathbb{E}_{q(\boldsymbol{\epsilon}|\mathbf{x}_t,c)}[\boldsymbol{\epsilon}|\mathbf{x}_t],\gamma_1\boldsymbol{\epsilon}_\theta(\mathbf{x}_t,c,t) + \gamma_0\boldsymbol{\epsilon}_\theta(\mathbf{x}_t,t)\rangle] \tag{S130}$$

$$= \mathbb{E}_{\mathbf{x}_0,\boldsymbol{\epsilon}}[\|\boldsymbol{\epsilon}\|_2^2 + \|\gamma_1\boldsymbol{\epsilon}_\theta(\mathbf{x}_t,c,t) + \gamma_0\boldsymbol{\epsilon}_\theta(\mathbf{x}_t,t)\|_2^2$$

$$- 2\mathbb{E}_{\mathbf{x}_0,\boldsymbol{\epsilon}}[\langle\boldsymbol{\epsilon}_\theta(\mathbf{x}_t,c,t),\gamma_1\boldsymbol{\epsilon}_\theta(\mathbf{x}_t,c,t) + \gamma_0\boldsymbol{\epsilon}_\theta(\mathbf{x}_t,t)\rangle] \tag{S131}$$

$$= \mathbb{E}_{\mathbf{x}_0,\boldsymbol{\epsilon}}[\|\boldsymbol{\epsilon}_\theta(\mathbf{x}_t,c,t)\|_2^2 + \|\gamma_1\boldsymbol{\epsilon}_\theta(\mathbf{x}_t,c,t) + \gamma_0\boldsymbol{\epsilon}_\theta(\mathbf{x}_t,t)\|_2^2$$

$$- 2\mathbb{E}_{\mathbf{x}_0,\boldsymbol{\epsilon}}[\langle\boldsymbol{\epsilon}_\theta(\mathbf{x}_t,c,t),\gamma_1\boldsymbol{\epsilon}_\theta(\mathbf{x}_t,c,t) + \gamma_0\boldsymbol{\epsilon}_\theta(\mathbf{x}_t)\rangle]$$

$$+ \mathbb{E}_{\mathbf{x}_0,\boldsymbol{\epsilon}}[\|\boldsymbol{\epsilon}\|_2^2 - \|\boldsymbol{\epsilon}_\theta(\mathbf{x}_t,c,t)\|_2^2] \tag{S132}$$

$$= \mathbb{E}_{\mathbf{x}_0,\boldsymbol{\epsilon}}[\|\boldsymbol{\epsilon}_\theta(\mathbf{x}_t,c,t) - (\gamma_1\boldsymbol{\epsilon}_\theta(\mathbf{x}_t,c,t) + \gamma_0\boldsymbol{\epsilon}_\theta(\mathbf{x}_t,t))\|_2^2] + C_2 \tag{S133}$$

$$= \mathbb{E}_{\mathbf{x}_0,\boldsymbol{\epsilon}}[\|(\gamma_1 - 1)\boldsymbol{\epsilon}_\theta(\mathbf{x}_t,c,t) + \gamma_0\boldsymbol{\epsilon}_\theta(\mathbf{x}_t,t)\|_2^2] + C_2, \tag{S134}$$

in which Eq. (S130) is from Lemma 1, and $C_2 = \mathbb{E}_{\mathbf{x}_0,\boldsymbol{\epsilon}}[\|\boldsymbol{\epsilon}\|_2^2 - \|\boldsymbol{\epsilon}_\theta(\mathbf{x}_t,c,t)\|_2^2]$ is constant not involving $\gamma_1$ and $\gamma_0$. As for $t = 1$ we have similar derivation:

$$\mathbb{E}_{q(\mathbf{x}_1,\mathbf{x}_0|c)}[-\log p_{\theta,\gamma_1,\gamma_0}(\mathbf{x}_0|\mathbf{x}_1,c))] \tag{S135}$$

$$\propto \mathbb{E}_{q(\mathbf{x}_1,\mathbf{x}_0|c)}[\|\mathbf{x}_0 - \mathbf{f}_{\theta,\gamma_1,\gamma_0}^t(\mathbf{x}_1,c)\|_2^2] + C_3 \tag{S136}$$

$$\propto \mathbb{E}_{\substack{\mathbf{x}_0\sim q(\mathbf{x}_0|c)\\\boldsymbol{\epsilon}\sim\mathcal{N}(\mathbf{0},\mathbf{I})\\\mathbf{x}_1=\alpha_1\mathbf{x}_0+\sigma_1\boldsymbol{\epsilon}}}[\|\boldsymbol{\epsilon} - (\gamma_1\boldsymbol{\epsilon}_\theta(\mathbf{x}_1,c,1) + \gamma_0\boldsymbol{\epsilon}_\theta(\mathbf{x}_1,1))\|_2^2] + C_4 \tag{S137}$$

$$= \mathbb{E}_{\mathbf{x}_0,\boldsymbol{\epsilon}}[\|(\gamma_1 - 1)\boldsymbol{\epsilon}_\theta(\mathbf{x}_1,c,1) + \gamma_0\boldsymbol{\epsilon}_\theta(\mathbf{x}_1,1))\|_2^2] + C_5, \tag{S138}$$

in which $C_3$, $C_4$, and $C_5$ are constants not involving $\gamma_1$ and $\gamma_0$. □

## B Pseudo-codes of Lookup Table

We below propose the pseudo-codes to achieve the lookup table and corresponding guided sampling in Algorithms 1 and 2.

**Algorithm 1** Pseudo-code to achieve lookup table of ReCFG in a PyTorch-like style.

```python
def calculate_lookup_table(net, gnet, data_loader, timesteps):
    """Defines the function to maintain the lookup table.

    Args:
        net: Noise prediction model for conditional score function.
        gnet: Noise prediction model for unconditional score function.
        data_loader: Dataloader to calculate score functions.
        timesteps: All timesteps under the given sampling trajectory.

    Returns:
        coeffs: Lookup list under all timesteps and conditions.
    """
    sum1_dict, sum2_dict = dict(), dict()
    # Iterate for the whole dataloader.
    for x, c in data_loader:
        # Iterate for all timesteps.
        sum1s, sum2s = list(), list()
        for nfe_idx, t in enumerate(timesteps):
            # Forward process.
            noise = torch.randn_like(x)
            x_t = alpha_t * x + sigma_t * noise

            # Calculate score functions first.
            eps_cond, eps_uncond = net(x_t, c, t), gnet(x_t, t)

            # Calculate the expectation in Eq. (34).
            sum1s.append(eps_cond.mean(dim=0, keepdim=True))
            sum2s.append(eps_uncond.mean(dim=0, keepdim=True))

        # Save the results.
        update_dict(sum1_dict, sum2_dict, c, sum1s, sum2s)

    # Calculate coefficients according to Eq. (34) for all timesteps.
    coeffs = {c: sum1_dict[c] / sum2_dict[c] for c in sum1_dict}

    return coeffs
```

**Algorithm 2** Pseudo-code for guided sampling by lookup table of ReCFG in a PyTorch-like style.

```python
def guided_sampler(sampler, net, gnet, gamma_1, noise, c, timesteps, coeffs):
    """Defines the guided sampling with lookup table.

    Args:
        sampler: Native sampler without guidance, e.g., DDIM sampler.
        net: Noise prediction model for conditional score function.
        gnet: Noise prediction model for unconditional score function.
        gamma_1: Guidance strength similar to CFG of type 'float'.
        noise: Initial random noise to denoise.
        c: Input label.
        timesteps: All timesteps under the given sampling trajectory.
        coeffs: Pre-calculated lookup table.

    Returns:
        x: A batch of samples by guided sampling.
    """
    # Calculate gamma_0.
    gamma_0s = (1. - gamma_1) * coeffs[c]
    # Ensure gamma_0 <= 0 and gamma_1 + gamma_0 >= 1.
    gamma_0s = clamp(gamma_0s, gamma_1)

    # Guided sampling using gamma_1 and gamma_0.
    x = noise
    for t, gamma_0 in zip(timesteps, gamma_0s):
        # Calculate score functions and apply guided sampling.
        eps_cond, eps_uncond = net(x, c, t), gnet(x, t)
        eps = eps_cond * gamma_1 + eps_uncond * gamma_0
        x = sampler(x, eps, t)

    return x
```

