# OpenReview forum: "Rectified Diffusion Guidance for Conditional Generation"
_ICLR.cc/2025/Conference — ICLR 2025 Conference Withdrawn Submission_

### Official Review · Reviewer_AeAF · 2024-10-19

**Soundness:** 3
**Presentation:** 2
**Contribution:** 2
**Rating:** 3
**Confidence:** 4

**Summary:**

The paper thoroughly examines Classifier-Free Guidance (CFG) in diffusion, highlighting the bias introduced by mean shift due to violations of the zero mean assumption in data samples. It suggests adjusting the CFG coefficient to mitigate this shift. Some empirical experiments seem to support these findings, revealing the mean-shift issue and proposing an easily implementable solution to address this bias.

**Strengths:**

The analysis of CFG is extensive.

**Weaknesses:**

* The proposed method appears numerically unstable due to the minute coefficient deviation from the default value of 1.0, casting doubt on the experiment's reliability. Furthermore, this slight difference in coefficients may indicate that the violation of the zero-mean assumption in the original CFG has negligible practical implications.  Please consider conducting an ablation study showing how results change with small perturbations to the coefficients.

* The merits of unbiasness scarfices the diversity(dramatic smaller variance compared to the ground truth), as indicated in Figure2. Provide a quantification of the trade-off between unbiasedness and diversity would be more informative.

* In essence, once relaxing the constriant of CFG: two coefficients sum to one; I doubt it increases or decreases the magnitude of the noise norm, which may collapse the sampling process. Providing an analysis of how the proposed method affects the noise norm throughout the sampling process, compared to standard CFG, would be informative.

* No visual demonstration is provided. Please provide some sample images generated with standard CFG vs. the proposed method, or visualizations of the sampling process over time.

**Questions:**

* Figure 1: Is the expectation ratio essentially the softmax? The figure's message is unclear. Distinguishing the expectation ratios between the first and last sampling images is challenging based on the comparison provided. Consider providing a more detailed explanation of what the expectation ratio represents and how it's calculated and some visualization, such as using a different color scale or providing separate figures for the first and last sampling steps.

* Figure2: Seems like this method achieves unbiasness at the cost of losing diversity(much smaller variance). If caclualting KL divergence, it is hard to tell which distribution is closer to the ground-truth.

* If relaxing the constriant of CFG: two coefficients sum to one; I doubt it increases or decreases the magnitude of the predicted noise norm, which may degenerate the sampling process.

* In Table1\&2: are the results of the baseline CFG good enough to be persuasive? Please choose some well-accepted benchmark models results on these datasets and ensure the hyperparameters are comparable, for example, 25 or 256 sampling steps with ddpm or ddim.

* From Table3, it suggests users have to prepare the expectation ratio for each class and each sampling steps. The combination of class and sampling steps makes it too cumbersome. Furthmore, the such small digit makes people worry the numerical stability of this method. Therefore, the ablation study should be presented to prove the numerical stability of the expectation ratio. Please consider conducting some ablation studies, such as:
Comparing results when using class-specific ratios vs. a global average ratio
Analyzing how performance changes when rounding the ratios to different decimal places
Examining the impact of using fewer sampling steps on both performance and computational complexity

---

### Official Review · Reviewer_TyAQ · 2024-10-29

**Soundness:** 4
**Presentation:** 3
**Contribution:** 3
**Rating:** 6
**Confidence:** 3

**Summary:**

This paper addresses conditional generative modeling and highlights the limitations of traditional Classifier-Free Guidance (CFG). It demonstrates that denoising with CFG cannot be viewed as a reciprocal diffusion process. To tackle this issue, the authors propose a relaxed version of CFG that introduces a new type of guidance. They show that this alternative guidance can be unbiased and provide a methodology for calculating the necessary guidance parameters. The paper also illustrates the superiority of their proposed method through evaluations on standard benchmarks, showcasing its effectiveness in overcoming the limitations of classic CFG.

**Strengths:**

- The paper is well written
- The proposed approach is novel
- The paper is well illustrated with intuitive theorems

**Weaknesses:**

- The proposed method doesn't give a closed form expression but an estimator
- The method is computationally intensive

**Questions:**

- It is difficult to understand the connection between theorems 1 and 2 as the fact that $\mathbb{E}\_{x_t}[\nabla_{x_t} \log q_{t, \gamma}(x_t \mid c)] \neq 0$ doesn't appear anywhere in theorem 2. However, theorem 3 illustrates this difference perfectly. Why putting theorem 3 so late in the manuscript ? I believe that putting theorem 3 early would help readers better undertsand the logical flow of the paper's intuitions.
- I find the results in Table 1 and 2 very tight between the algorithms, could you provide means and standard deviations ?
- According to Table 3, it seems that the expectation ratio is often close to 1. Doesn't it advocate for $\gamma_0 = 1 - \gamma_1$ in the end ?

---

### Official Review · Reviewer_vH2L · 2024-10-31

**Soundness:** 3
**Presentation:** 3
**Contribution:** 3
**Rating:** 5
**Confidence:** 2

**Summary:**

The paper examines rectified classifier-free guidance (ReCFG) for diffusion models, addressing theoretical flaws in existing models that use Classifier-Free Guidance (CFG). By introducing new coefficients for combining conditional and unconditional score functions, ReCFG adheres closely to diffusion theory, eliminating expectation shifts in conditional distributions and enhancing model fidelity without the need for retraining.

**Strengths:**

1. The paper provides a theoretical analysis, identifying a flaw in the popular CFG technique and proposing a mathematically grounded solution, also provides rigorous proof of the benefits of ReCFG.
2. The introduction of ReCFG offers a novel solution to the expectation shift problem associated with CFG in diffusion models.

**Weaknesses:**

1. Since it is mainly a theoretical paper, I think it would be beneficial to add more explanation after Theorem 3, while probably reduce the length of Theorem 2.
2. If I understand correctly, the method needs to solve (28) for each time step t, which may increase the computation.
3. Although it is a theoretical paper, it would be better to add more experiment results, e.g., some illustrations of generated images.

**Questions:**

1. I guess one needs to solve (28) for each time step t, which may introduce more computation. Will that be an issue?
2. Whether the coefficients obtained by optimizing (37) can lead to (28) since it also take expectations with respect to t?
3. By (28) and the equation before (34), it seems that we also have $E_{x_t}(\epsilon_\theta(x_t,t,c)) = E_{x_t}(\epsilon_\theta(x_t,t)) = 0$. Now suppose you have $\Delta_t =0$. From the proof of Theorem 3, specifically (S107) and (S112), we have $E_{x_0}(x_0) = E_{c,x_0}(x_0)$. Does that mean in this case, the conditional expectation is the same as the unconditional expectation? If it is true, does that mean if $E_{x_0}(x_0) \neq  E_{c,x_0}(x_0)$, $\Delta_t \neq 0$ and you will always have expectation shift?

---

### Official Review · Reviewer_En9m · 2024-11-04

**Soundness:** 2
**Presentation:** 2
**Contribution:** 2
**Rating:** 3
**Confidence:** 2

**Summary:**

This paper introduces an enhancement to Classifier-Free Guidance (CFG), a widely used approach in diffusion model sampling, by identifying and addressing an expectation shift flaw in CFG. The authors propose "Rectified Classifier-Free Guidance" (ReCFG), a variation that relaxes the original guidance coefficient constraints to align more closely with diffusion theory. By incorporating a lookup table to adjust guidance coefficients, ReCFG aims to maintain conditional fidelity without additional retraining or loss of inference speed. The authors present theoretical derivations and empirical results demonstrating improved conditional fidelity on several state-of-the-art diffusion models, including applications on ImageNet and CC12M.

**Strengths:**

The paper provides a rigorous analysis of a known issue with CFG, quantifying expectation shift and proposing ReCFG as a theoretically grounded alternative. This analysis contributes to understanding the mathematical foundation of guided sampling in diffusion models. The resulting ReCFG is designed as a post-hoc solution that can be applied to pretrained models without the need for additional training.

The experiments on the toy data is helpful for understanding the method and subsequent real data experiments also demonstrates the effectiveness.

**Weaknesses:**

My main concern is the scope of the experiment and demonstrated experimental improvements. CFG has been a very important component in most AIGC model. Even GPT-type sequential image modeling uses CFG for better performance. Targeting such a component, the current work doesn't demonstrate enough practical impact in more complex generative tasks such as text to image generation.
Performance-wise, the empirical gain measured by the FID or CLIP scores in ImageNet and CC12M are not that significant. In this case, some user study or visual comparison would be helpful in showcasing the effectiveness.

**Questions:**

1. The unbiasedness is a great property to have. However, from the toy case experiment, the proposed method seems to affect the variance recovery. Would the proposed method hurt diversity in image generation?

2. In text-to-3D generation, e.g., DreamFusion, the CFG coefficient is a lot larger. Can the analysis explain this and can similar improvements made in these scenarios?

3. While ReCFG’s use of a lookup table aids in theoretical correctness, it adds a layer of complexity to the sampling process. For text-to-image models, where open-vocabulary synthesis requires flexibility across diverse prompts, the need to precompute or interpolate values for novel prompts could hinder ReCFG’s feasibility. I am wondering how much is the extra computation, especially for more complicated generation tasks.

---

### Note · Authors · 2024-11-12

I have read and agree with the venue's withdrawal policy on behalf of myself and my co-authors.